# Impaired ABCA1/ABCG1-mediated lipid efflux in the mouse retinal pigment epithelium (RPE) leads to retinal degeneration

Federica Storti[1], Katrin Klee[1,2], Vyara Todorova[1,3], Regula Steiner[4], Alaa Othman[4], Saskia van der Velde-Visser[5], Marijana Samardzija[1], Isabelle Meneau[6], Maya Barben[1], Duygu Karademir[1,2], Valda Pauzuolyte[1†], Sanford L Boye[7], Frank Blaser[6], Christoph Ullmer[8], Joshua L Dunaief[9], Thorsten Hornemann[4], Lucia Rohrer[4], Anneke den Hollander[5,10], Arnold von Eckardstein[4], Jürgen Fingerle[11], Cyrille Maugeais[8], Christian Grimm[1,2,3]*

[1]Lab for Retinal Cell Biology, Department of Ophthalmology, University of Zurich, Schlieren, Switzerland; [2]Center for Integrative Human Physiology, University of Zurich, Zurich, Switzerland; [3]Neuroscience Center Zurich, University of Zurich, Zurich, Switzerland; [4]Institute of Clinical Chemistry, University of Zurich, Schlieren, Switzerland; [5]Department of Human Genetics, Radboud University Medical Center, Nijmegen, Netherlands; [6]Department of Ophthalmology, University Hospital Zurich, Zurich, Switzerland; [7]Department of Ophthalmology, University of Florida, Gainesville, United States; [8]Roche Pharma Research and Early Development, Roche Innovation Center Basel, F Hoffmann-La Roche Ltd., Basel, Switzerland; [9]Department of Ophthalmology, Scheie Eye Institute, University of Pennsylvania, Philadelphia, United States; [10]Department of Ophthalmology, Radboud University Medical Center, Nijmegen, Netherlands; [11]Natural and Medical Sciences Institute, University of Tübingen, Tübingen, Germany

*For correspondence:
cgrimm@opht.uzh.ch

Present address: †Stem Cells and Regenerative Medicine Section, UCL Great Ormond Street Institute of Child Health, London, United Kingdom

**Abstract** Age-related macular degeneration (AMD) is a progressive disease of the retinal pigment epithelium (RPE) and the retina leading to loss of central vision. Polymorphisms in genes involved in lipid metabolism, including the ATP-binding cassette transporter A1 (*ABCA1*), have been associated with AMD risk. However, the significance of retinal lipid handling for AMD pathogenesis remains elusive. Here, we study the contribution of lipid efflux in the RPE by generating a mouse model lacking ABCA1 and its partner ABCG1 specifically in this layer. Mutant mice show lipid accumulation in the RPE, reduced RPE and retinal function, retinal inflammation and RPE/photoreceptor degeneration. Data from human cell lines indicate that the *ABCA1* AMD risk-conferring allele decreases *ABCA1* expression, identifying the potential molecular cause that underlies the genetic risk for AMD. Our results highlight the essential homeostatic role for lipid efflux in the RPE and suggest a pathogenic contribution of reduced ABCA1 function to AMD.
DOI: https://doi.org/10.7554/eLife.45100.001

## Introduction

Age-related macular degeneration (AMD) is the leading cause of blindness in the elderly population of Western countries (*Klein et al., 2013*; *Joachim et al., 2015*) and its socio-economic impact is

predicted to dramatically increase in the next decades (*Wong et al., 2014*). AMD is a progressive disease of the macula, the central cone-rich region of the retina, and can develop into the 'dry' or 'wet' form in the advanced stage. Dry AMD is characterized by atrophy of the retinal pigment epithelium (RPE) and photoreceptor degeneration, while wet AMD exhibits pathological neo-vascularization of the retina originating from the choroid. Both conditions eventually result in loss of RPE and photoreceptors with deleterious consequences on high acuity and color vision (*Bird et al., 1995*; *Lim et al., 2012*).

The etiology of AMD is complex and multifactorial but several lines of evidence associate the disease with local disturbances of lipid metabolism in the ageing human eye (*Pauleikhoff et al., 1990*). Lipids physiologically accumulate in extracellular deposits known as drusen and sub-retinal drusenoid deposits (SDDs) on the basal and apical side of the RPE, respectively. Drusen contain polar lipids, such as un-esterified (free) cholesterol (UC) and phosphatidylcholine (PC), as well as neutral lipids, such as cholesteryl esters (CEs), and several lipid-binding proteins (apolipoproteins) (*Wang et al., 2010*; *Curcio et al., 2011*). The more recently identified SDDs, instead, seem to contain UC only, together with apolipoproteins (*Rudolf et al., 2008*; *Spaide et al., 2018*). Drusen (*Sarks, 1980*) and SDDs (*Zweifel et al., 2010*) are considered hallmarks of AMD but their actual origin and contribution to the pathology remain unknown. Recently, primary RPE cells isolated from AMD patients, but not from control subjects, were shown to accumulate intracellular lipids *in vitro* (*Golestaneh et al., 2017*), suggesting altered lipid metabolism in diseased cells.

Genome-wide association studies have linked AMD to several genes involved in generation and remodeling of high-density lipoproteins (HDLs), namely ATP-binding cassette transporter A1 (*ABCA1*), apolipoprotein E (*APOE*), cholesteryl ester transfer protein (*CETP*) and hepatic lipase C (*LIPC*) (*Fritsche et al., 2016*). A recent review (*van Leeuwen et al., 2018*) summarizes contradictory results from different studies concerning the association between systemic lipid levels and the risk of developing AMD and links long-term elevated plasma levels of HDL-cholesterol to increased AMD risk. However, it remains unknown whether genes involved in lipid metabolism exert a local and/or a systemic pathogenic effect on the retina.

A gene of interest in this context is *ABCA1*, encoding a transmembrane lipid transporter which generates HDLs together with its partner ABCG1. Either transporter uses ATP to flip lipids, mainly UC and phospholipids (PLs), but also sphingomyelins (SMs) and oxysterols, from the inner leaflet of the plasma membrane to extracellular lipophilic acceptors such as apolipoproteins or nascent HDLs. ABCA1 initiates the formation of HDL by direct interaction with naked apolipoproteins, while ABCG1 requires a lipidated particle (*Cavelier et al., 2006*; *Quazi and Molday, 2011*; *Li et al., 2013*). Since UC is one of the best established substrates of the two transporters, the ABCA1/ABCG1 pathway is also known as 'active cholesterol efflux'. The fundamental role of this pathway for cellular lipid homeostasis is highlighted by macrophage foam cell formation (*Tabas, 2002*; *Favari et al., 2015*) and by the progressive and age-dependent lung phenotype, including lipid accumulation in alveolar macrophages and pneumocytes, lung dysfunction and inflammation (*Chai et al., 2017*), in mice lacking ABCA1, ABCG1 or both. Inhibition of cholesterol efflux leads to cell dysfunction also in pancreatic beta cells (*Kruit et al., 2012*), neurons (*Karasinska et al., 2013*) and liver cells (*Arguello et al., 2015*). Liver X receptors (LXR) α and β are the upstream regulators of the pathway: these two transcription factors are activated upon binding of oxysterols that accumulate in conditions of increased UC and upregulate expression of both *ABCA1* and *ABCG1* (*Schultz et al., 2000*).

Ubiquitous expression of ABCA1 and ABCG1 has been reported in the mouse, monkey and human retina, including the RPE (*Tserentsoodol et al., 2006*; *Duncan et al., 2009*; *Zheng et al., 2012*; *Ananth et al., 2014*; *Zheng et al., 2015*; *Storti et al., 2017*). The function of ABCA1 and ABCG1 in the RPE was previously investigated *in vitro* (*Ishida et al., 2006*; *Duncan et al., 2009*; *Biswas et al., 2017*; *Storti et al., 2017*; *Lyssenko et al., 2018*) and shown to mediate transport of UC to ApoA-I, ApoE, HDLs and human serum on both sides of the RPE. This was true for plasma lipoprotein- as well as outer segment (OS)-derived cholesterol. However, the relevance of active cholesterol efflux for the RPE *in vivo* remains unknown. This, together with the fact that the RPE needs an efficient metabolism to handle large amounts of lipids coming from daily OS phagocytosis (*Strauss, 2005*), prompted us to generate an RPE-specific *Abca1;Abcg1* double knockout (KO) mouse. We characterize the retinal phenotype of this mouse model and provide evidence

suggesting a correlation between AMD-associated *ABCA1* genotypes and expression levels of this gene in human cells.

## Results

### Generation of RPE-specific *Abca1;Abcg1* double KO mice (RPE$^{\Delta Abca1;Abcg1}$)

Expression of ABCA1 and ABCG1 throughout the retinal layers, including the RPE, was confirmed by immunofluorescence (IF) in wild type mouse retinal sections (*Figure 1A*) (*Ananth et al., 2014*). As previously described for RPE cells *in vitro* (*Storti et al., 2017*), no co-localization with ezrin (EZR), a marker of the apical microvilli of the RPE, was observed. In order to study the function of ABCA1/ABCG1 in the RPE, we used *BEST1Cre* mice to delete floxed sequences from *Abca1$^{flox/flox}$;Abcg1-$^{flox/flox}$* mice and generate RPE-specific *Abca1;Abcg1* double KOs (called RPE$^{\Delta Abca1;Abcg1}$, see 'Materials and methods', *Table 1* and *Figure 1—figure supplement 1*). *BEST1Cre* mice express *Cre* recombinase under control of the human bestrophin 1 (*BEST1*, also known as vitelliform macular dystrophy 2, *VMD2*) promoter, resulting in post-natal CRE activity specifically in the RPE (*Iacovelli et al., 2011*). Although both strains were used before to successfully generate a number of mouse models (*Westerterp et al., 2012*; *Yao et al., 2015*; *Westerterp et al., 2016*; *Sundermeier et al., 2017*; *Ban et al., 2018a*; *Ban et al., 2018b*; *Eblimit et al., 2018*; *Roman et al., 2018*), we nonetheless validated the specificity of *Cre* expression in RPE$^{\Delta Abca1;Abcg1}$ mice. High mRNA levels for *Cre* were detected in the eyecup (RPE/choroid) with only a minimal amount of transcripts found in the neural retina, probably due to contamination during eye dissection (*Figure 1B*). To confirm presence of CRE protein in the RPE, we performed IF staining on retinal sections. Although some un-specific staining was observed in the inner retina, CRE-positive nuclei were detected only in the RPE layer of RPE$^{\Delta Abca1;Abcg1}$ but not of control (Ctr, *Cre*-negative) mice (*Figure 1C*). Finally, we checked for successful CRE-mediated excision of floxed fragments by amplifying *Abca1* and *Abcg1* specific sequences from genomic DNA extracted from retina and eyecups (including RPE) of RPE$^{\Delta Abca1;Abcg1}$ and Ctr mice. As expected, deletion of *Abca1* and *Abcg1* was observed in eyecups, but not neural retinas, of *Cre*-positive mice (*Figure 1D*). Even though end-point PCR reactions may not be used to quantify products, the highly variable signal intensities of the amplified *Abca1* and *Abcg1* excised fragments suggested mouse-to-mouse variability in *Cre* expression (*Figure 1B* and data not shown) and/or in deletion efficiency (*Figure 1D*). Of note, the *BEST1Cre* mouse is known to have patchy and variable *Cre* expression in the RPE (*Iacovelli et al., 2011*; *Sundermeier et al., 2017*), which could partially explain decreased rather than abolished expression of *Abca1* and *Abcg1* mRNA in eyecups of RPE$^{\Delta Abca1;Abcg1}$ mice (*Figure 1—figure supplement 2*).

### Lack of *Abca1* and *Abcg1* in the RPE leads to morphological alterations and intracellular lipid accumulation

Already at 2 months of age, the fundus of RPE$^{\Delta Abca1;Abcg1}$ but not of Ctr mice showed a dotted pattern, possibly reflecting alterations in the pigmentation of RPE cells (*Figure 2A*). Light and electron microscopy on retinal sections revealed an irregular apical RPE border and accumulation of intracellular material resembling lipid droplets (LDs) in RPE$^{\Delta Abca1;Abcg1}$ but not Ctr mice (*Figure 2B and C*). Staining with OilRedO (ORO) in retinal sections (*Figure 2D*) and LipidTOX in RPE flat mounts (*Figure 2E*) revealed strong lipid accumulation in CRE-positive RPE cells of RPE$^{\Delta Abca1;Abcg1}$ mice. CRE-negative RPE cells and other retinal layers were ORO-negative and served as internal controls demonstrating the specificity of lipid accumulation in RPE cells lacking *Abca1* and/or *Abcg1*. Both ORO and LipidTOX stain neutral lipids, which constitute the hydrophobic core of LDs (*Olofsson et al., 2009*). Moreover, actin staining of RPE flat mounts showed morphological irregularities of CRE-positive cells in RPE$^{\Delta Abca1;Abcg1}$ mice when compared to the regular, mainly hexagonal shape of Ctr cells (*Figure 2F*). These morphological irregularities progressively worsened and were more pronounced at 4–6 months of age (*Figure 3*). In particular, double staining for the tight junction protein zona occludens 1 (ZO-1) and the Wnt signaling mediator β-catenin (β-cat) in mutant cells revealed re-localization of β-cat from the plasma membrane to the cytosol, a feature of disorganized RPE (*Yang et al., 2018*). Pigment epithelial cells in 4 months old RPE$^{\Delta Abca1;Abcg1}$ mice were

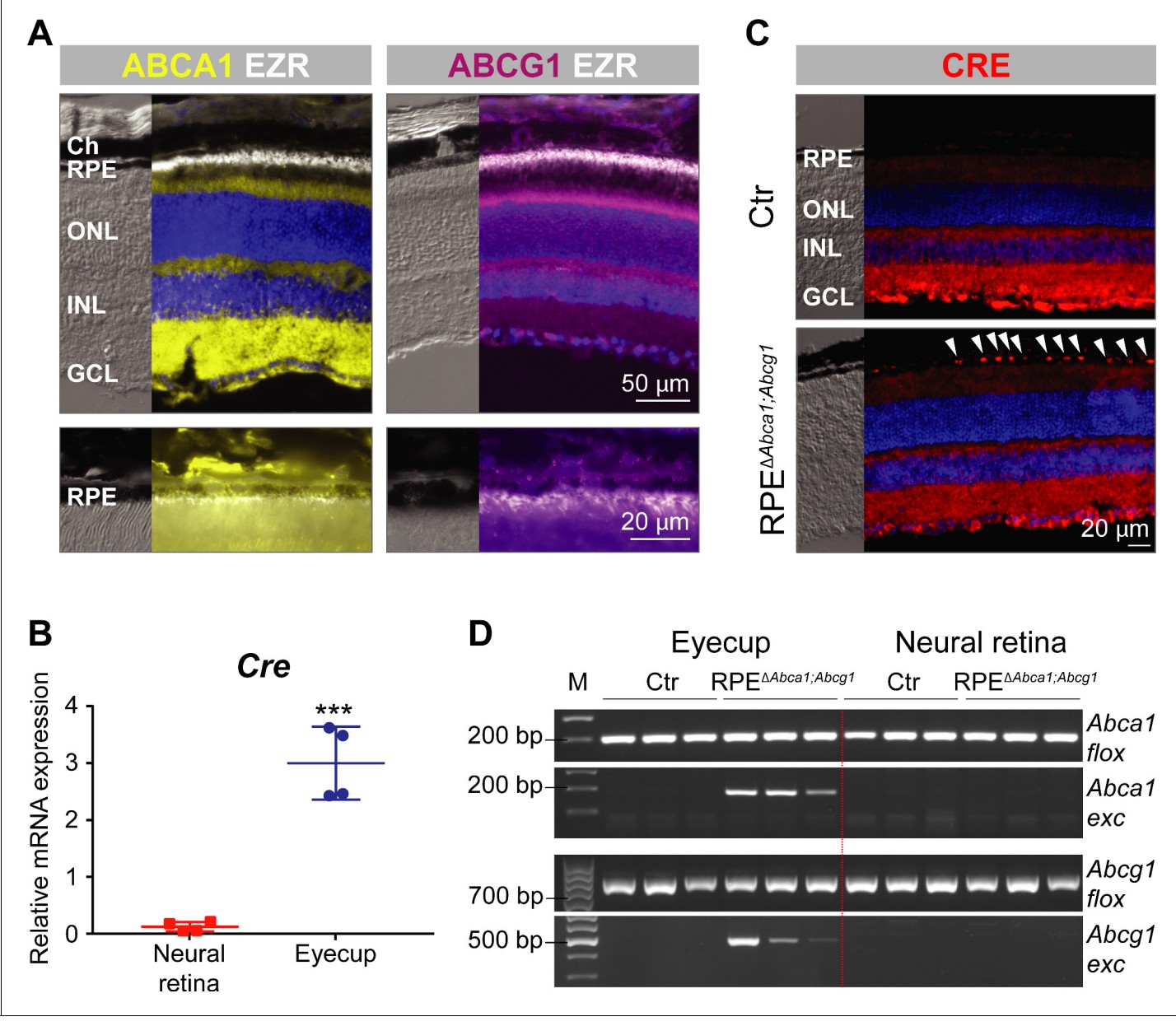

**Figure 1.** Generation of RPE$^{\Delta Abca1;Abcg1}$ mice. (**A**) IF staining for ABCA1 (yellow), ABCG1 (violet) and the RPE apical marker EZR (white) in retinas of 2-months-old wt mice. Lower panels show magnification of the RPE layer. Nuclei were counterstained with DAPI (blue). Ch: choroid; RPE: retinal pigment epithelium; ONL: outer nuclear layer; INL: inner nuclear layer; GCL: ganglion cell layer. (**B**) *Cre* mRNA levels were measured by semi-quantitative real-time PCR in neural retinas and eyecups (RPE/choroid) from 2-months-old RPE$^{\Delta Abca1;Abcg1}$ mice. Shown are data from individual samples and means ± standard deviations (SD, N = 4). Statistics: Student's t-test; ***: p<0.001. (**C**) IF staining for CRE (red) in retinal sections from 2-months-old Ctr and RPE$^{\Delta Abca1;Abcg1}$ mice: white arrowheads indicate CRE-positive nuclei in the RPE of mutant mice. Nuclei were counterstained with DAPI (blue). Note the non-specific signal in the inner retina. Representative pictures of N = 6 mice. (**D**) Detection of CRE-mediated excision fragments in *Abca1* and *Abcg1* (*Abca1/Abcg1 exc*) by conventional PCR on genomic DNA from eyecups and neural retinas of Ctr and RPE$^{\Delta Abca1;Abcg1}$ mice (N = 3). For this picture, animals showing heterozygous deletion of *Abca1/Abcg1* in ear biopsies (see 'Materials and methods') were excluded in order to detect excision truly due to CRE expression in the eye. PCR for the floxed sequences (*Abca1/Abcg1 flox*) was performed as positive control. Shown are PCR products run on a 2% agarose gel and visualized with ethidium bromide. Note the lack of the excised fragment in the neural retina. M: DNA size marker, indicated fragment sizes are shown in base pairs (bp).

DOI: https://doi.org/10.7554/eLife.45100.002

The following figure supplements are available for figure 1:

**Figure supplement 1.** Genotyping and definition of the mouse models.
DOI: https://doi.org/10.7554/eLife.45100.003

*Figure 1 continued*

**Figure supplement 2.** Gene expression in eyecups of Ctr and RPE$^{\Delta Abca1;Abcg1}$ mice.

DOI: https://doi.org/10.7554/eLife.45100.004

significantly larger and irregularly shaped (*Figure 3A*, quantification in 3B and 3C). At 6 months, we observed areas of dysmorphic RPE with accumulation of intracellular material and areas of RPE atrophy with infiltration of inflammatory cells (see below) in mutant but not control mice (*Figure 3D and E*). Photoreceptor loss correlated with RPE atrophy (see below). Variability of the phenotype within the same retina was probably due to patchy *Cre* expression (*Figures 1* and *2*). Reduced expression levels of *Cre* and the RPE marker monocarboxylic acid transporter 3 (*Mct3*) further indicated atrophic RPE at 6 months of age (*Figure 3F and G*). Loss of RPE cells in aged RPE$^{\Delta Abca1;Abcg1}$ mice was most likely a consequence of the lack of ABCA1 and/or ABCG1 activity in RPE rather than of CRE expression per se, since *Cre* mRNA levels declined in eyecups of old RPE$^{\Delta Abca1;Abcg1}$ (*Figure 3F*) but not old *BEST1Cre* mice (*Figure 3—figure supplement 1*). Similarly, morphological abnormalities of RPE cells in RPE$^{\Delta Abca1;Abcg1}$ mice were not due to potential CRE toxicity (*Thanos et al., 2012*; *He et al., 2014*) as *BEST1Cre* mice only showed minor morphological alterations in the RPE and few bright spots in the fundus, but no lipid accumulation or functional changes (*Figure 3—figure supplement 2*). Thus, lack of *Abca1* and/or *Abcg1* resulted in lipid accumulation in the RPE and led to several morphological abnormalities that aggravated with time and eventually resulted in RPE cell death.

To exclude developmental effects as a cause for the phenotype, we tested lipid accumulation in the RPE after inactivation of *Abca1* and *Abcg1* in adult mice. For this purpose, we injected an adeno-associated virus (AAV) expressing *Cre* and green fluorescent protein (*GFP*) under the control of the *BEST1* promoter (*Figure 4A*) into the sub-retinal space of adult *Abca1$^{flox/flox}$;Abcg1$^{flox/flox}$* mice. Although expression levels of GFP were variable and difficult to detect in some individual cells, LDs were specifically observed in GFP-positive (transduced) cells by LipidTOX staining 10 weeks after AAV injection (*Figure 4B*). In addition, RPE cells in the transduced area appeared larger and less regular than in the non-transduced area, similar to the morphological alterations detected in RPE$^{\Delta Abca1;Abcg1}$ mice (*Figure 2F*). Lipid accumulation in GFP-positive RPE of *Abca1$^{flox/flox}$;Abcg1$^{flox/flox}$* mice was further confirmed in retinal sections, which showed ORO-positive lipid staining specifically in the RPE of transduced areas, as well as co-localization of GFP and CRE signals (*Figure 4C–F*). Contralateral eyes were injected with phosphate buffer saline (PBS, vehicle control) to check for any injection-related effects and showed, as expected, no lipid accumulation (not shown). Taken

**Table 1.** Mice genotypes and nomenclature.
*flox/-*: detection of floxed and excised (KO) allele in ear biopsy.

| Genotype | Name |
|---|---|
| *Abca1$^{flox/flox}$;Abcg1$^{flox/flox}$* | *Cre*-negative controls: Ctr |
| *Abca1$^{flox/-}$;Abcg1$^{flox/flox}$* | |
| *Abca1$^{flox/flox}$;Abcg1$^{flox/-}$* | |
| *Abca1$^{flox/-}$;Abcg1$^{flox/-}$* | |
| *Abca1$^{flox/flox}$;Abcg1$^{flox/flox}$;BEST1Cre* | RPE-specific double KOs: RPE$^{\Delta Abca1;Abcg1}$ |
| *Abca1$^{flox/-}$;Abcg1$^{flox/flox}$;BEST1Cre* | |
| *Abca1$^{flox/flox}$;Abcg1$^{flox/-}$;BEST1Cre* | |
| *Abca1$^{flox/-}$;Abcg1$^{flox/-}$;BEST1Cre* | |
| *Abca1$^{flox/flox}$;Abcg1$^{+/+}$;BEST1Cre* | RPE-specific *Abca1* single KOs: RPE$^{\Delta Abca1}$ |
| *Abca1$^{flox/-}$;Abcg1$^{+/+}$;BEST1Cre* | |
| *Abca1$^{+/+}$;Abcg1$^{flox/flox}$;BEST1Cre* | RPE-specific *Abcg1* single KOs: RPE$^{\Delta Abcg1}$ |
| *Abca1$^{+/+}$;Abcg1$^{flox/-}$;BEST1Cre* | |
| *Abca1$^{+/+}$;Abcg1$^{+/+}$;BEST1Cre* | *Cre*-positive controls: BEST1Cre |

DOI: https://doi.org/10.7554/eLife.45100.005

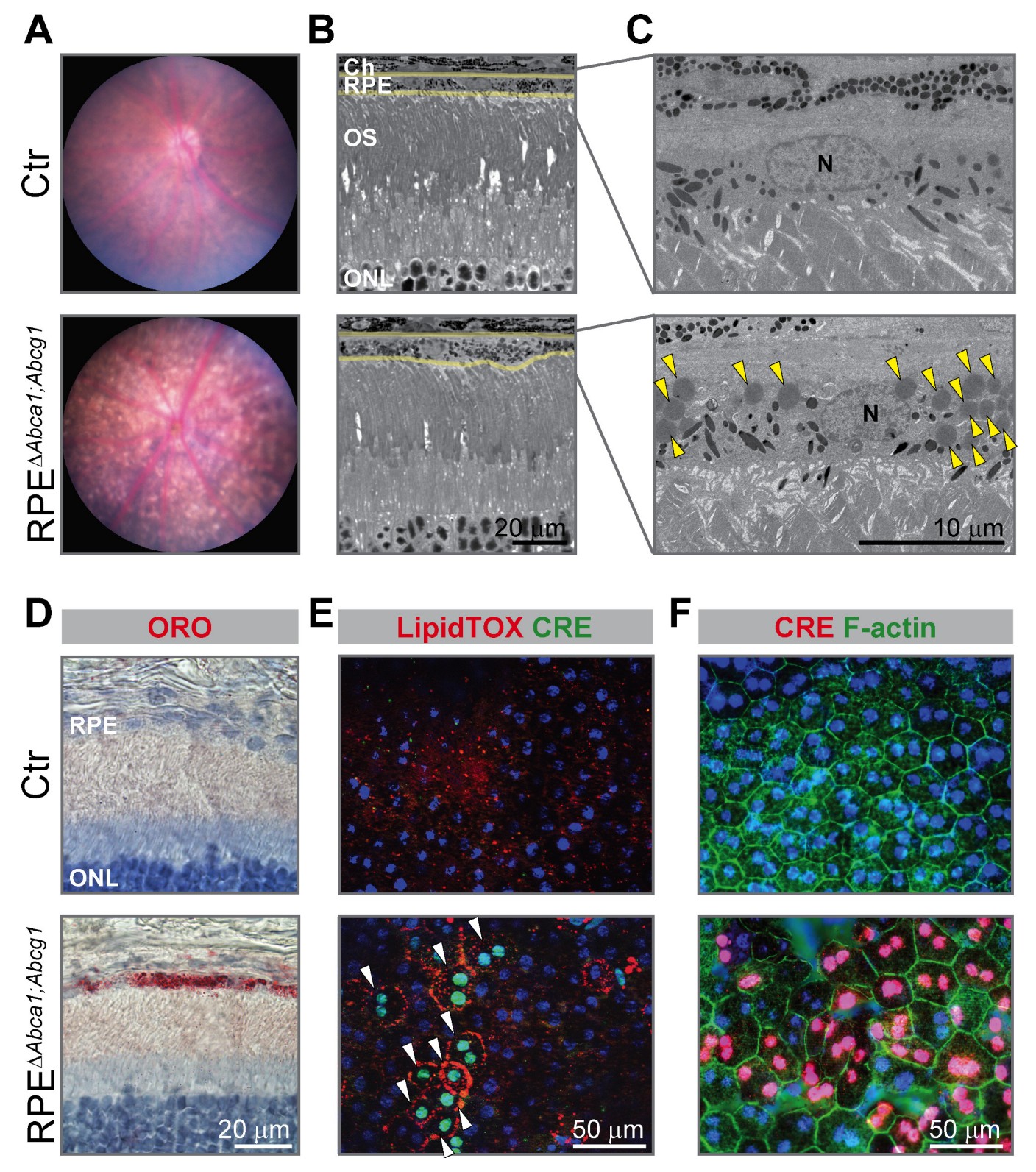

**Figure 2.** Early morphological alterations and intracellular lipid accumulation in RPE$^{\Delta Abca1;Abcg1}$ mice. (A) Fundus imaging of 2-months-old Ctr and RPE$^{\Delta Abca1;Abcg1}$ mice showing altered pigmentation pattern in mutant mice. Corresponding retinal morphology analyzed by light (B) and electron (C) microscopy revealed alterations of the RPE in RPE$^{\Delta Abca1;Abcg1}$ mice. Yellow lines in (B) indicate RPE borders. Yellow arrowheads in (C) indicate lipid droplets. OS: outer segments; N: nucleus. (D) Retinal sections were stained with ORO (red, dye for neutral lipids); nuclei were counterstained with

*Figure 2 continued on next page*

*Figure 2 continued*

hematoxylin (blue). RPE flat mounts were stained with LipidTOX (red, dye for neutral lipids) and anti-CRE (green) (E) or anti-CRE (red) and phalloidin (green, staining actin filaments) (F). Nuclei were counterstained with Hoechst. White arrowheads indicate CRE-positive cells showing lipid accumulation in mutant mice. Representative pictures of N ≥ 3 animals per group. Abbreviations as in *Figure 1*.

DOI: https://doi.org/10.7554/eLife.45100.006

together, these data indicated altered morphology and intracellular lipid accumulation in adult RPE cells lacking *Abca1* and/or *Abcg1*. This phenotype is in agreement with the known function of ABCA1/ABCG1 as mediators of lipid efflux in the RPE.

## Lipid droplets in the RPE of RPE$^{\Delta Abca1;Abcg1}$ mice are composed mainly of cholesteryl esters

We next characterized the lipid composition of eyecups from 2-months-old RPE$^{\Delta Abca1;Abcg1}$ and Ctr mice. We performed the same analysis on the corresponding neural retinas in order to evaluate possible effects of impaired lipid transport in the RPE on lipid homeostasis of other retinal cells. Additionally, plasma samples from the same mice were included to check for presence of any systemic changes on circulating lipid levels that could contribute to the eye phenotype. We used mass spectrometry-based approaches to measure a broad number of lipid classes and species. The analysis revealed significantly increased concentration of CEs in eyecups of RPE$^{\Delta Abca1;Abcg1}$ mice. In contrast UC, PLs, sphingolipids (SLs) including sphingomyelins (SMs) and ceramides (Cer), and glycerolipids (GLs) including diacylglycerols (DAGs) and triglycerides (TGs) remained unchanged (*Figure 5A*). All of the individual CE species analyzed were more abundant in the mutant mice compared to control littermates. Some CE species were dramatically increased up to 100 fold (*Figure 5B*), including CEs containing fatty acid chains typically found in the retina such as palmitic (16:0), oleic (18:1) and docosahexaenoic (22:6) acid, which is the most abundant fatty acid of photoreceptor OS (*Fliesler and Anderson, 1983*; *Martin et al., 2005*; *Bretillon et al., 2008*). No major difference in the lipid composition was detected in the neural retinas of the two strains (*Figure 5C*), apart from a modest but significant increase in CE levels. However, the small extent of the increase and the low concentration (2.7 ± 1.1 pmol/µg protein in the neural retina, 1239.9 ± 955.1 pmol/µg protein in the eyecup, *Supplementary file 1A*) suggested a contamination from the RPE during tissue dissection rather than a real increase in the neural retina. Systemic lipid levels measured in the plasma showed no differences between RPE$^{\Delta Abca1;Abcg1}$ and Ctr mice in any of the considered classes (*Figure 5D*), supporting a local effect of the lack of *Abca1* and *Abcg1* in the RPE. The high variability observed in plasma lipid levels might be explained by the fact that the mice had access to food *ad libitum*, thus, in our experiment, lipid intake was uncontrolled. Analysis of lipid composition therefore revealed prominent accumulation of CEs in the RPE of RPE$^{\Delta Abca1;Abcg1}$ mice without major alterations of the neural retina or plasma lipidomes. Absolute concentrations for all of the analyzed lipid classes can be found in *Supplementary file 1A*. Finally, we also detected a significant increase in the relative abundance of the visual cycle intermediates retinyl esters (REs) in eyecups of mutant mice (*Figure 5E*).

## Functional consequence of lipid accumulation in the RPE

Since RPE$^{\Delta Abca1;Abcg1}$ mice revealed alterations in RPE morphology and lipid composition, we tested whether lack of *Abca1* and *Abcg1* affected function of the epithelium. For this purpose, we investigated rhodopsin (RHO) regeneration kinetics after bleaching, a major task of the RPE in the classical visual cycle (*Strauss, 2005*). 2-months-old RPE$^{\Delta Abca1;Abcg1}$ and Ctr mice had similar dark levels of RHO, which were bleached with comparable efficiencies (*Supplementary file 1B*). However, RPE$^{\Delta Abca1;Abcg1}$ mice regenerated RHO slower within the first 30 min after bleaching. After this initial phase, the amount of regenerated RHO was no longer different between the mice (*Figure 6* and *Supplementary file 1B*). This suggests an early delay in the visual cycle, probably due to difficulties with handling the incoming all-*trans* retinol.

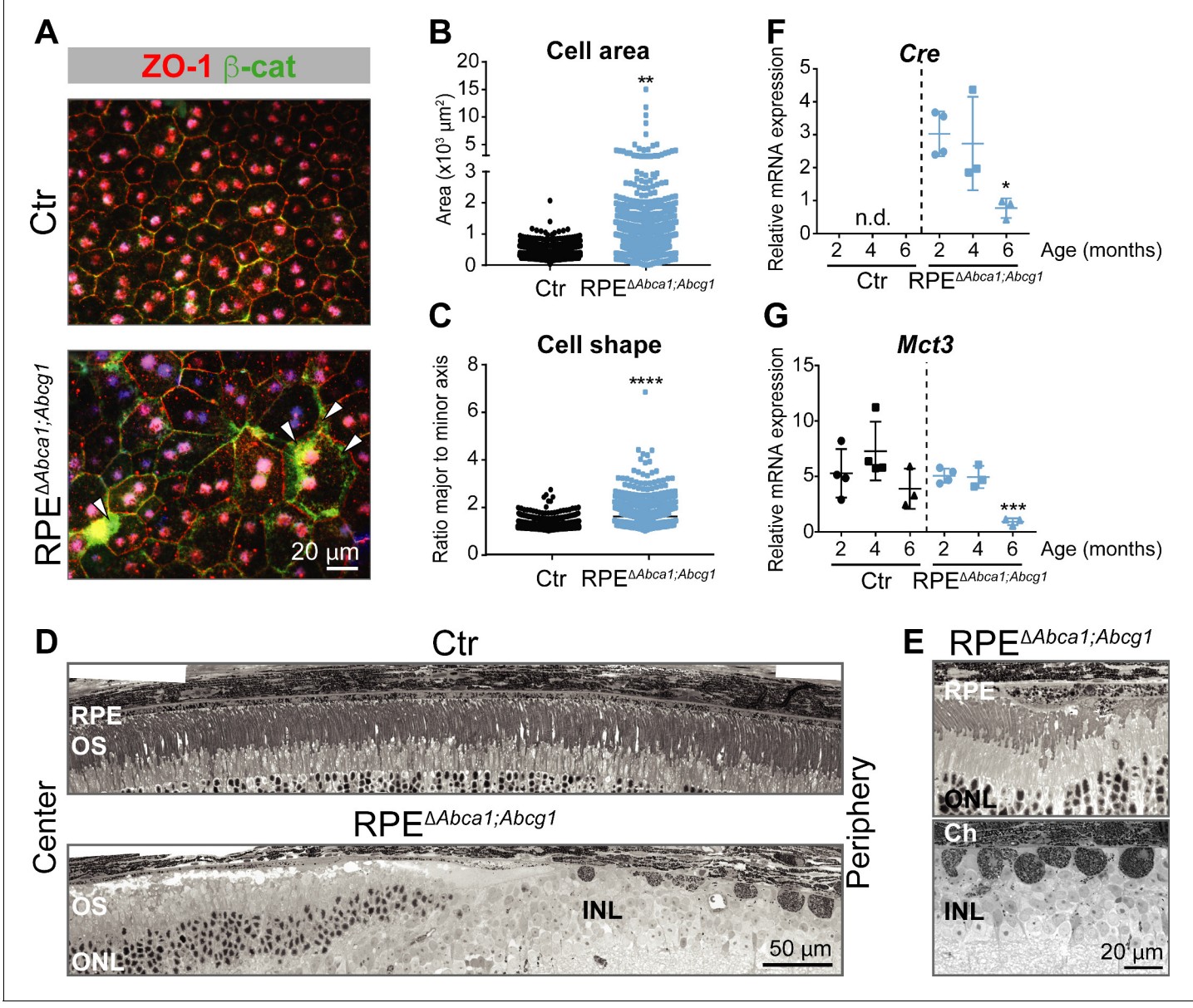

**Figure 3.** Effect of lipid accumulation in the ageing mouse RPE. (**A**) RPE flat mounts from 4-months-old Ctr and RPE$^{\Delta Abca1;Abcg1}$ mice were stained for ZO-1 (red) and β-cat (green). White arrowheads indicate loss of co-localization between ZO-1 and β-cat in mutant RPE. Nuclei were counterstained with Hoechst. Shown are representative images of N = 3 animals per group. Quantification of cell area (**B**) and cell shape (**C**) was performed using ImageJ on images from ZO-1 stained flat mounts. Corresponding measurements of single analyzed cell can be found in *Figure 3—source data 1*. Statistics: Mann-Whitney test; **: p<0.01, ****: p<0.0001. Light microscopy was used to visualize outer retinas of control and RPE$^{\Delta Abca1;Abcg1}$ mice: shown are panoramas (**D**) and RPE at higher magnification (**E**). Representative images of N ≥ 3 animals per group. *Cre* (**F**) and *Mct3* (**G**) mRNA levels were measured by semi-quantitative real-time PCR in eyecups from Ctr and RPE$^{\Delta Abca1;Abcg1}$ mice at the indicated ages. Shown are data from individual samples and means ± SD (N = 3–4). Statistics: one-way ANOVA vs '2 months' of the respective genotype; *: p<0.05, ***: p<0.001. n.d.: not detected. Abbreviations as in *Figure 1*.

DOI: https://doi.org/10.7554/eLife.45100.007

The following source data and figure supplements are available for figure 3:

**Source data 1.** RPE cell Area and shape of RPE cells in RPE$^{\Delta Abca1;Abcg1}$ and control mice.
DOI: https://doi.org/10.7554/eLife.45100.010

**Figure supplement 1.** *Cre* expression in RPE$^{\Delta Abca1;Abcg1}$ and *BEST1Cre* mice.
DOI: https://doi.org/10.7554/eLife.45100.008

**Figure supplement 2.** Absence of retinal phenotype in *BEST1Cre* mice up to 6 months of age.
DOI: https://doi.org/10.7554/eLife.45100.009

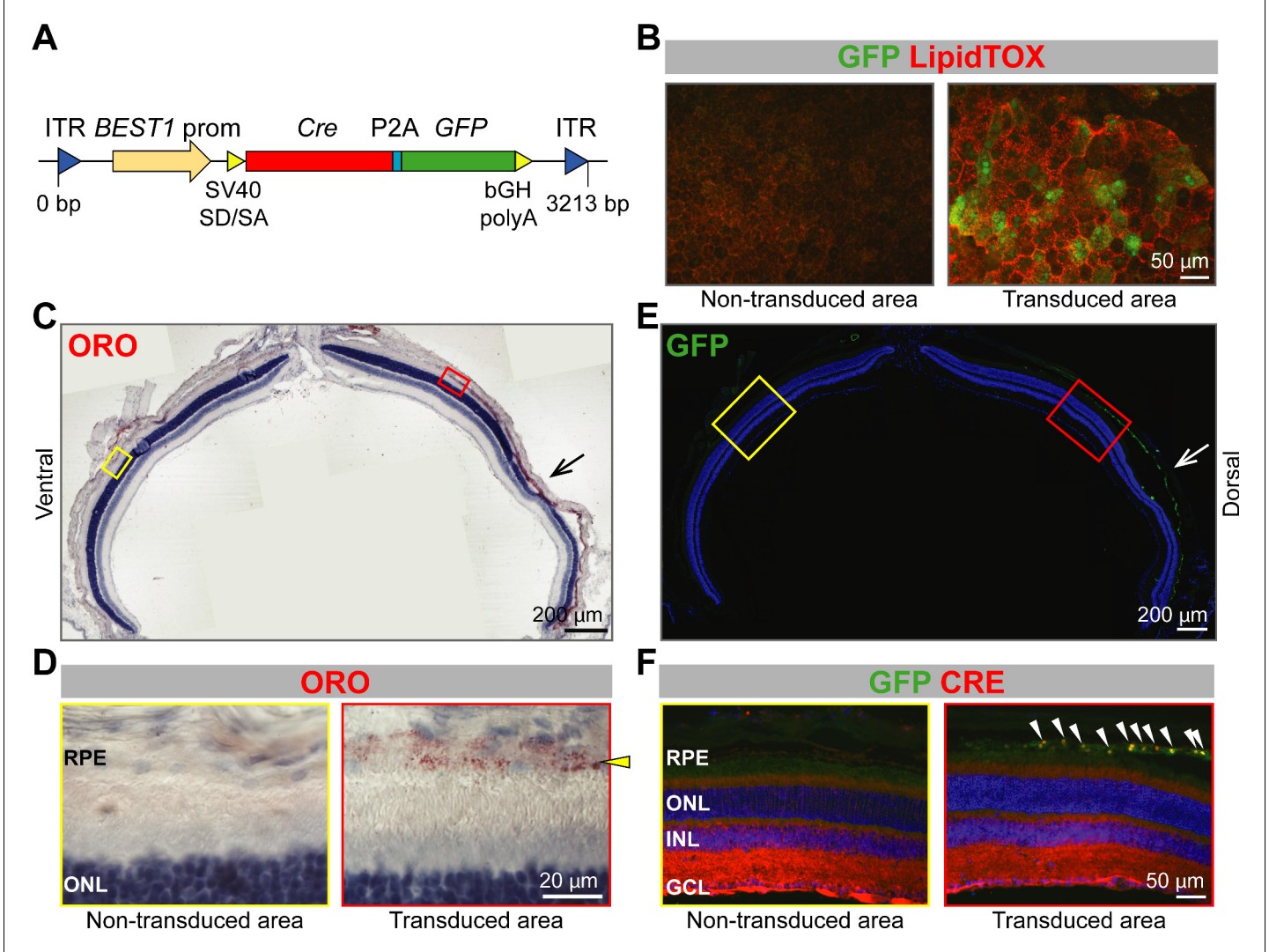

**Figure 4.** Lipid accumulation after AAV-mediated excision of *Abca1* and *Abcg1* in adult RPE. (**A**) Schematic representation of the vector packaged into AAV4 capsid in order to express *Cre* and *GFP* specifically in the RPE of *Abca1$^{flox/flox}$;Abcg1$^{flox/flox}$* mice. Length of the construct in base pairs is shown below the map. ITR: inverted terminal repeat; SV40 SD/SA: simian virus 40 splice donor/splice acceptor site; P2A: porcine teschovirus 2A; bGH polyA: bovine growth hormone polyadenylation tail. 10 weeks after sub-retinal injections, co-localization of AAV-mediated *Cre/GFP* expression and lipid accumulation was analyzed by IF in RPE flat mounts (**B**) and retinal sections (**C–F**). (**B**) RPE flat mounts were stained with LipidTOX (red); shown are representative images of a non-transduced and a transduced area. Dorsal-ventral retinal sections were stained with ORO: retina panorama is shown in (**C**) and magnified images of a non-transduced and a transduced area (corresponding to yellow and red rectangles in the panorama) are shown in (**D**). Yellow arrowhead indicates LDs in the transduced RPE. Nuclei were counterstained with hematoxylin (blue). Consecutive retinal sections were analyzed for AAV transduction by IF: retinal panorama is shown in (**E**) and magnified pictures of a non-transduced and a transduced area (corresponding to yellow and red rectangles in the panorama) are shown in (**F**), together with CRE staining. White arrowheads indicate CRE-positive nuclei in the transduced RPE. Nuclei were counterstained with DAPI (blue). Black (**C**) and white (**E**) arrows indicate the injection site. Representative pictures of N ≥ 3 animals per group. Abbreviations as in *Figure 1*.

DOI: https://doi.org/10.7554/eLife.45100.011

## Lack of *Abca1* and *Abcg1* in the RPE results in age-dependent retinal degeneration

Loss of ABCA1 and ABCG1 from mouse RPE resulted in early lipid accumulation, morphological alterations and atrophy of this cellular layer. To understand the consequences of such diseased RPE for the neural retina, we imaged the mutant mice at different ages (2, 4 and 6 months) by fundus photography and optical coherence tomography (OCT). The pigmentation changes observed in

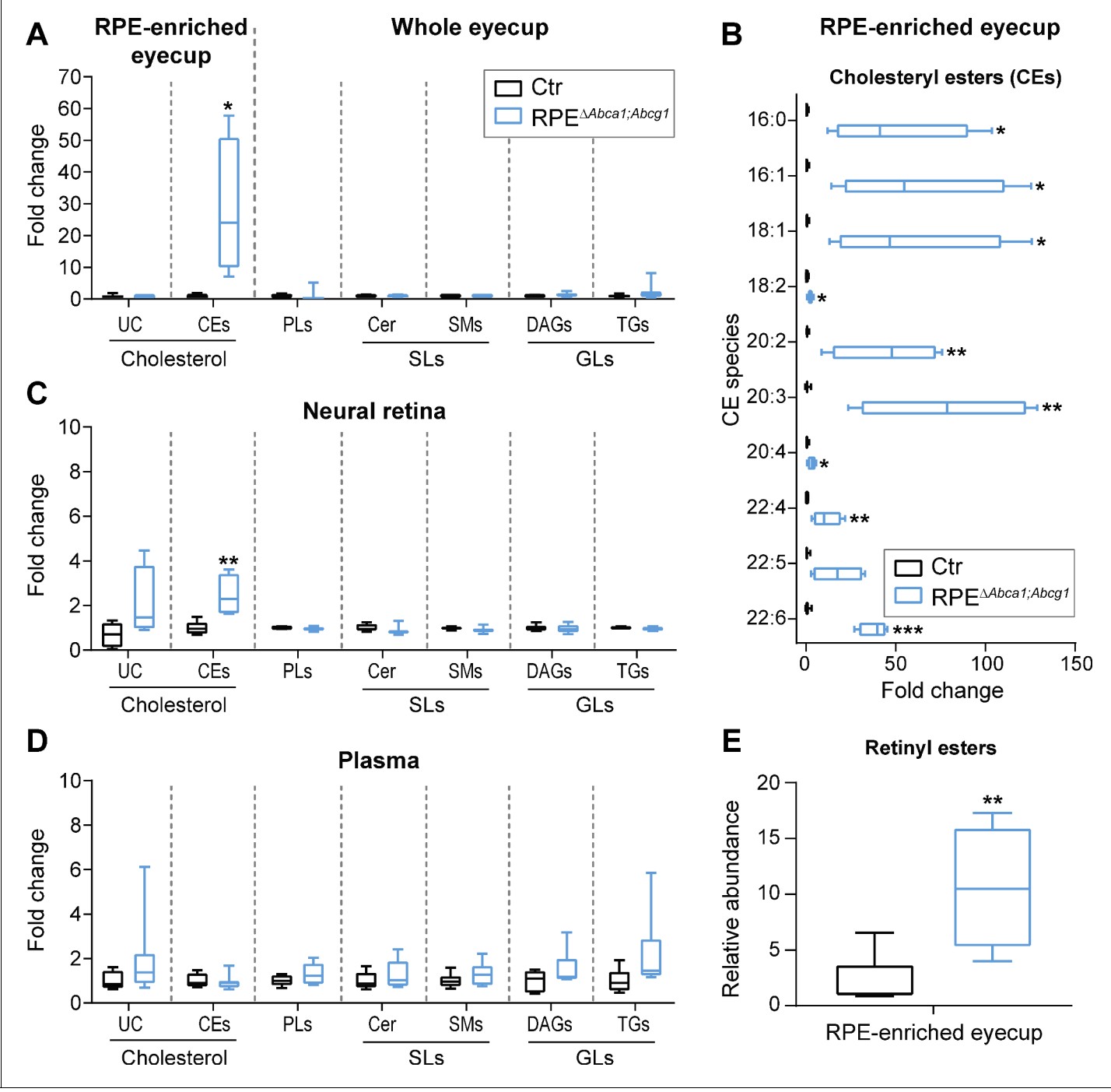

**Figure 5.** Cholesteryl esters as main components of LDs in the RPE of RPE$^{\Delta Abca1;Abcg1}$ mice. Lipid composition of eyecups (**A**), neural retinas (**C**) and plasma (**D**) from 2-months-old Ctr and mutant mice was measured by mass spectrometry-based methods. The following lipid classes were analyzed: cholesterol (un-esterified cholesterol, UC, and cholesteryl esters, CEs), phospholipids (PLs: sum of phosphatidylcholine, phosphatidylethanolamine, phosphatidylserine, phosphatidylinositol and phosphatidylglycerol), sphingolipids (SLs: ceramides, Cer, and sphingomyelins, SMs) and glycerolipids (GLs: diacylglycerols, DAGs, and triglycerides, TGs). (**B**) Cholesteryl esters species containing the indicated fatty acids were quantified in eyecups from the same animals. (**E**) Relative quantification of retinyl esters was performed in eyecups from 2-months-old Ctr and RPE$^{\Delta Abca1;Abcg1}$ mice. Shown are box plots of folds on respective Ctr average, whiskers correspond to min and max values (N = 4–10). Lipid concentration values corresponding to fold changes in (**A**), (**C**) and (**D**) as well as single PL classes can be found in **Supplementary file 1A**. Please note that UC, CEs and REs were determined in RPE-enriched eyecups whereas PLs, SLs and GLs were determined in whole eyecups. Also, tissues from both eyes of the same animals were used for analysis of UC, CEs and REs, whereas tissues from single eyes were used for PLs, SLs and GLs (see 'Materials and methods'). Statistics: Student's t-test vs 'Ctr'; *: p<0.05, **: p<0.01, ***: p<0.001.

DOI: https://doi.org/10.7554/eLife.45100.012

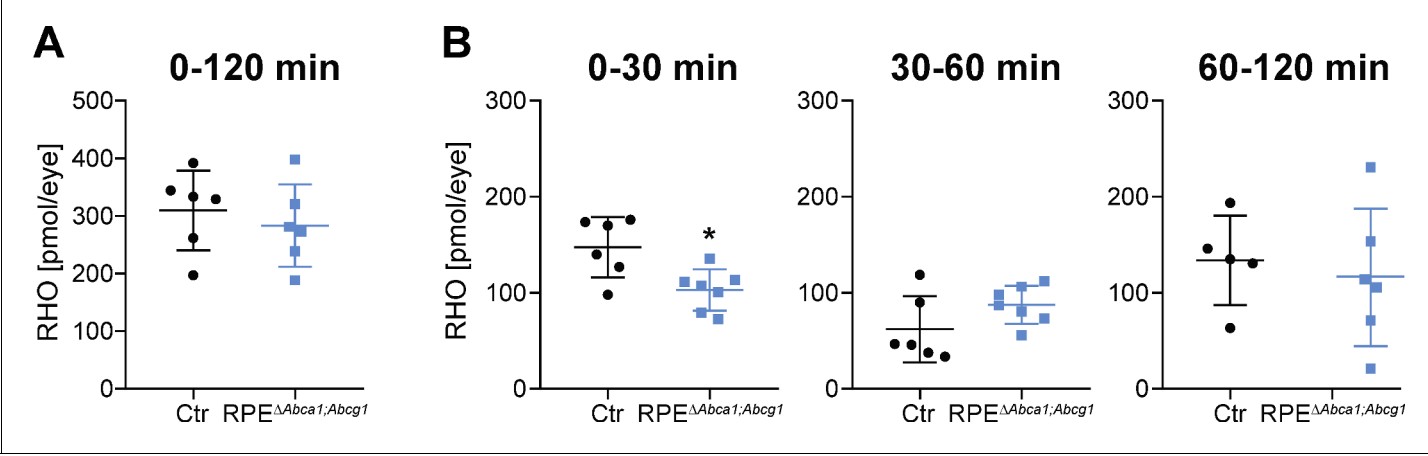

**Figure 6.** Delayed RHO regeneration in RPE$^{\Delta Abca1;Abcg1}$ mice. Dark-adapted 2-months-old Ctr and RPE$^{\Delta Abca1;Abcg1}$ mice were exposed to 5'000 lux for 10 min and the RHO content was measured in each retina. Dark controls were kept in darkness for the entire procedure. RHO levels were measured in dark controls, immediately after bleach (0 min) and after 30, 60 and 120 min of recovery in darkness. (**A**) 'Total' amount of regenerated RHO after 120 min was calculated by subtracting the corresponding averaged RHO amount at '0 min' from the RHO levels at '120 min'. (**B**) Amount of regenerated RHO during the indicated time intervals after bleaching were calculated by subtracting the corresponding averaged RHO amount at the early time point from the RHO levels measured at the later time point. Shown are data from individual samples and means ± SD (N = 4–8 eyes, corresponding to 2–4 mice). Statistics: Student's t-test vs 'Ctr'; *: p<0.05. Averages and SD of RHO content measurements can be found in *Supplementary file 1B*; single measurements per eye can be found in *Figure 6—source data 1*.

DOI: https://doi.org/10.7554/eLife.45100.013

The following source data is available for figure 6:

**Source data 1.** Rhodopsin regeneration in RPE$^{\Delta Abca1;Abcg1}$ and control mice.

DOI: https://doi.org/10.7554/eLife.45100.014

RPE$^{\Delta Abca1;Abcg1}$ mice at two months of age (*Figure 2A*) worsened at older ages (*Figure 7A*). OCT scans revealed sub-retinal hyper-reflective foci in mutant mice starting at 4 months of age (*Figure 7A*). These foci were accompanied by irregular RPE/outer nuclear layer (ONL) borders and retinal thinning, suggesting ongoing degeneration. Analysis of the respective retinal morphologies (*Figure 7B*) confirmed degenerative processes in the RPE/photoreceptor layers in ageing RPE$^{\Delta Abca1;Abcg1}$ mice. Retinal degeneration was further supported by a significant reduction of the ONL thickness in mutant vs control mice at 6 months of age (*Figure 7C*). The high variability in ONL measurements was likely owed to the patchy expression of the *Cre* transgene resulting in areas with intact RPE/ONL and areas with RPE cell death and consequent photoreceptor degeneration within the same retinal section. In some regions, both RPE and ONL were completely lost (see also *Figure 3D*). The inner retina was instead not affected by ABCA1/ABCG1 knockout in the RPE, as revealed by the determination of the INL thickness and staining for ganglion cells in 6-months-old animals (*Figure 7—figure supplement 1*).

Progressing photoreceptor degeneration was also reflected by the retinal function measured by electroretinography (ERG). Scotopic and photopic wave amplitudes gradually decreased in ageing RPE$^{\Delta Abca1;Abcg1}$ mice starting already at 4 months of age (*Figure 8*). In conclusion, lack of *Abca1* and *Abcg1* in the RPE had a strong impact on neural retinal morphology and function, with progressive photoreceptor degeneration.

## Inflammatory response in RPE$^{\Delta Abca1;Abcg1}$ mice

Retinal sections analyzed by light microscopy suggested the presence of immune cells in aged mutant mice (*Figure 3D and E*) and infiltration of inflammatory cells in the retina is one of the key events in AMD pathogenesis (*Kauppinen et al., 2016*). We thus stained RPE flat mounts and retinal sections of RPE$^{\Delta Abca1;Abcg1}$ mice for macrophages/activated microglia markers. At 4 months of age, up to about 100 ionized calcium-binding adapter molecule 1 (IBA-1)-positive cells were detected in flat mounts of all mutant RPE at the sites of morphological alterations, but not in non-affected areas (not shown) or Ctr mice. Confocal microscopy showed that IBA-1-positive signals were located within

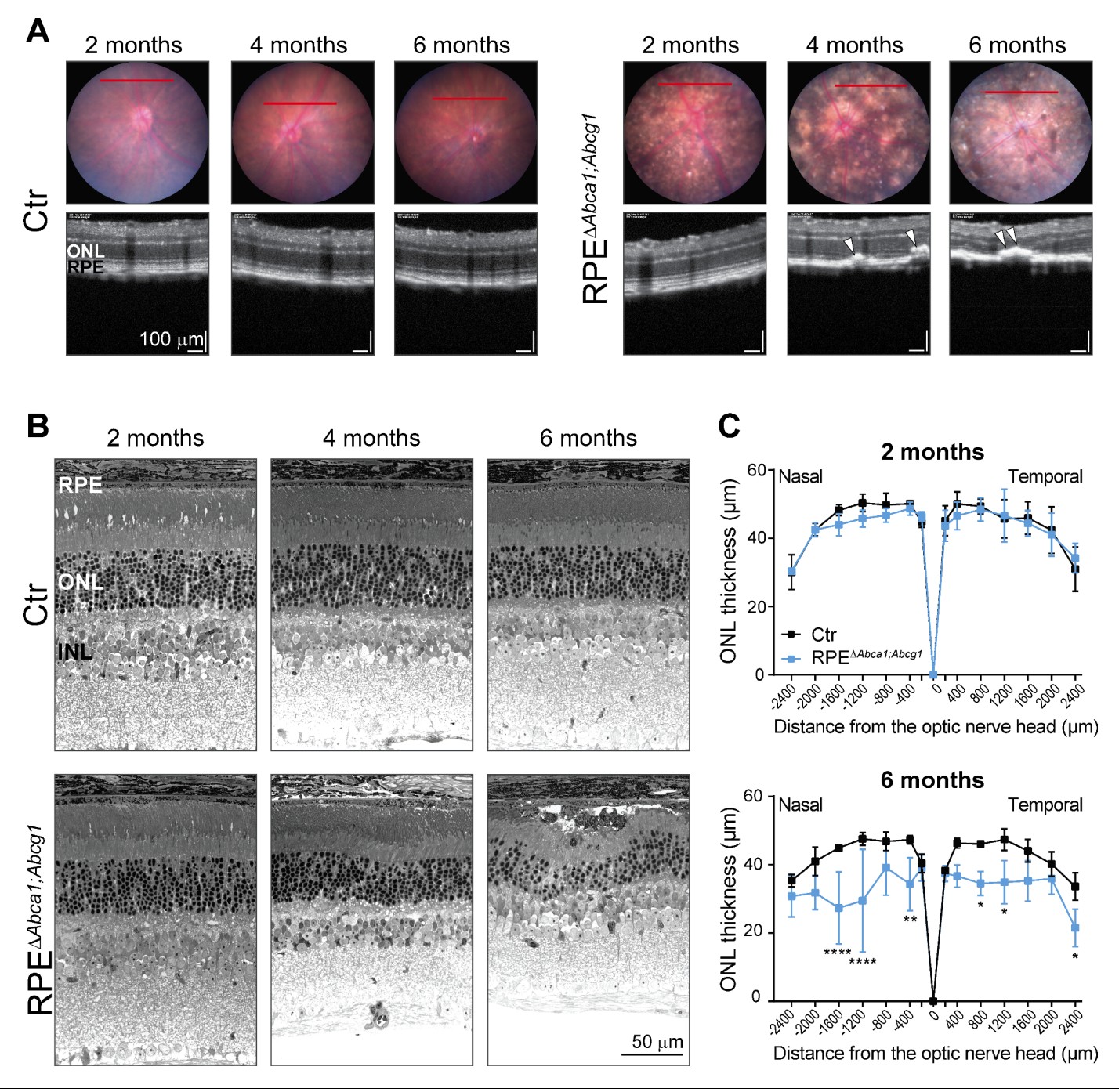

**Figure 7.** Age-dependent retinal degeneration in RPE$^{\Delta Abca1;Abcg1}$ mice. (**A**) Fundus images (upper panels) and OCT scans (lower panels, corresponding to red lines in fundus) of Ctr and RPE$^{\Delta Abca1;Abcg1}$ mice at the indicated age. White arrowheads indicate sub-retinal hyper-reflective foci. Retinal morphology of the same animals was analyzed by light microscopy (**B**). Representative pictures of N $\geq$ 3 animals per group. ONL thickness was quantified from nasal-temporal panorama images at 2 and 6 months of age and presented as spidergrams (**C**): significant ONL thinning was detected in 6-months-old RPE$^{\Delta Abca1;Abcg1}$ mice. Shown are means $\pm$ SD (N $\geq$ 3). Statistics: two-way ANOVA with Sidak's multiple comparison test; *: p<0.05, **: p<0.01, ****: p<0.0001. Abbreviations as in *Figure 1*.

DOI: https://doi.org/10.7554/eLife.45100.015

The following figure supplement is available for figure 7:

**Figure supplement 1.** Absence of a phenotype in the inner retina of RPE$^{\Delta Abca1;Abcg1}$ mice at 6 months of age.

DOI: https://doi.org/10.7554/eLife.45100.016

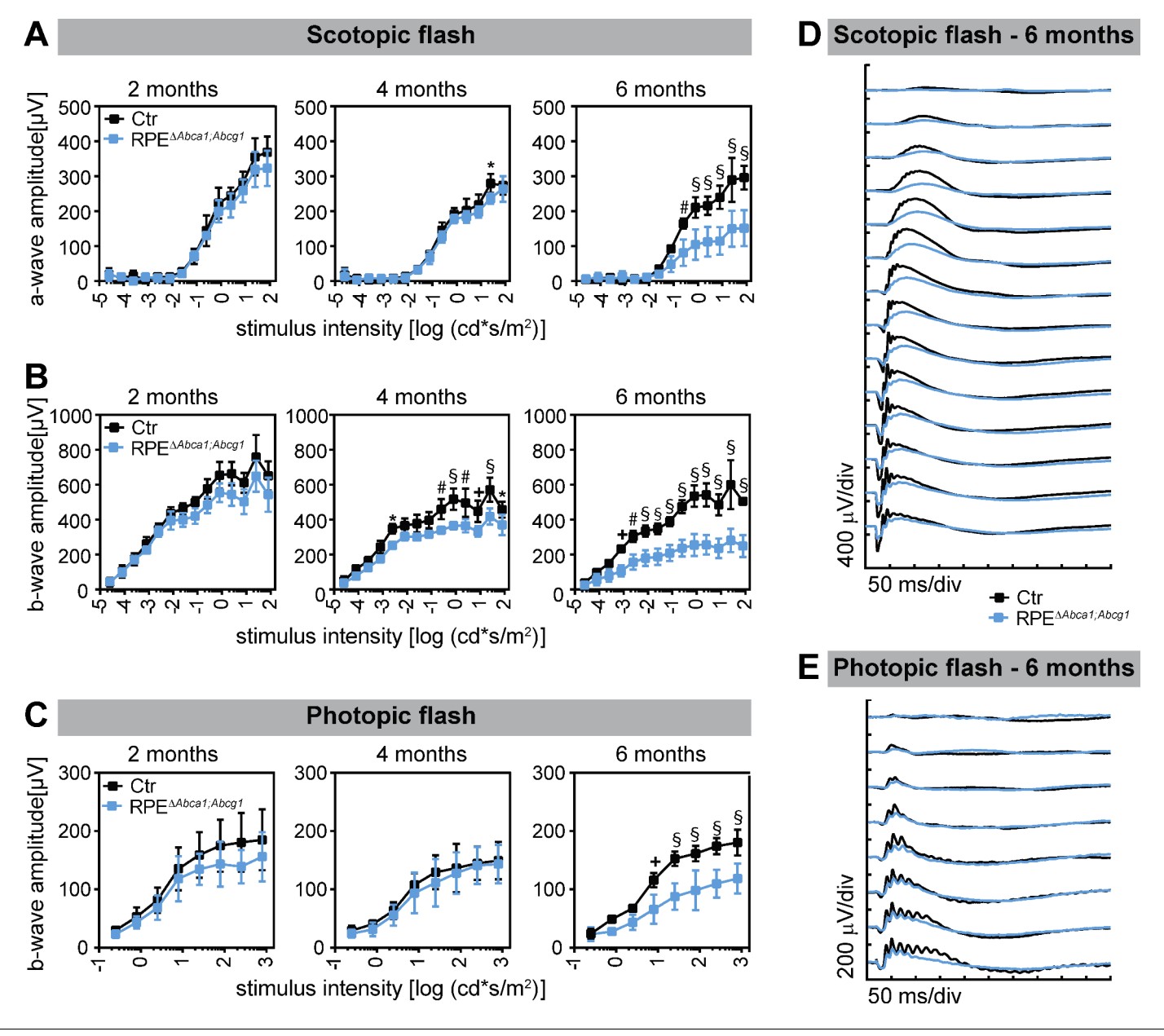

**Figure 8.** Decreased retinal function in aged RPE$^{\Delta Abca1;Abcg1}$ mice. Scotopic and photopic ERGs were recorded with increasing light intensities from dark-adapted Ctr and RPE$^{\Delta Abca1;Abcg1}$ mice at the indicated ages. Shown are mean ± SD (N = 3–6) of scotopic a- (**A**) and b-wave (**B**) amplitudes as well as photopic b-wave (**C**) amplitudes. Average scotopic and photopic traces of 6-months-old animals are shown in (**D**) and (**E**), respectively. Statistics: two-way ANOVA with Sidak's multiple comparison test; *: p<0.05, +: p<0.01, #: p<0.001, §: p<0.0001.
DOI: https://doi.org/10.7554/eLife.45100.017

the RPE layer as well as on its basal side (*Figure 9A*, lower cross-sections). Whether they represent cells infiltrating the RPE from the choroidal (basal) side or leaving the retina through the RPE from the apical side was not determined. IBA-1 positive inflammatory cells were also detected in the outer retinal layers including the sub-retinal space of 6 months old RPE$^{\Delta Abca1;Abcg1}$ but not control mice (*Figure 9B*). At this later time point, such cells were not only present in regions of strong photoreceptor and RPE atrophy (not shown, but see *Figure 3D,E* and *Figure 7B* for retinal morphologies showing large, presumably inflammatory cells in the sub-retinal space) but also in retinal regions that were mildly affected (*Figure 9B*). It is conceivable that damaged RPE cells facilitated the movement

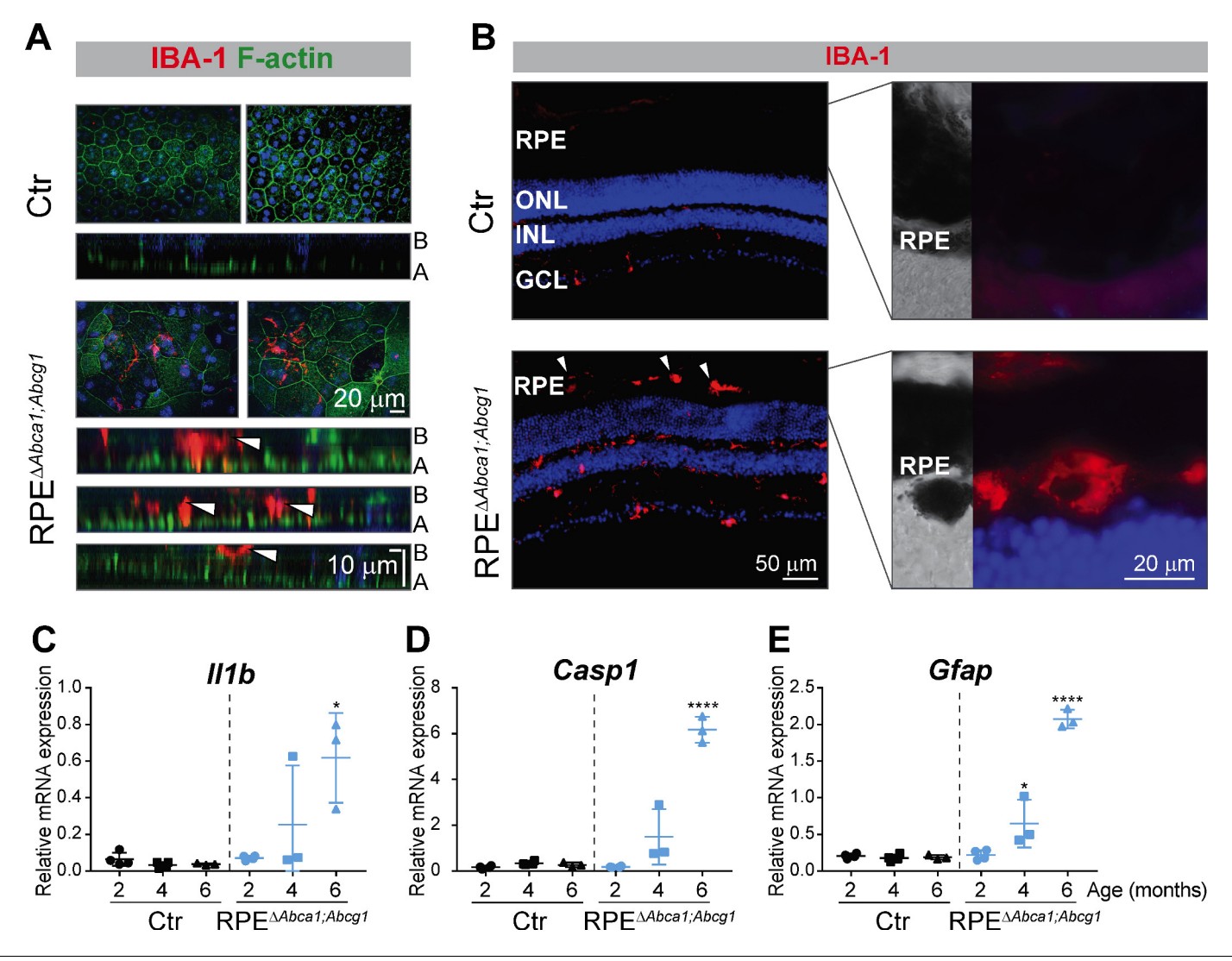

**Figure 9.** Inflammatory response in RPE$^{\Delta Abca1;Abcg1}$ mice. (**A**) RPE flat mounts from 4-months-old Ctr and RPE$^{\Delta Abca1;Abcg1}$ mice were stained with phalloidin (green, staining actin filaments) and anti-IBA-1 (red). Shown are representative top-view images and cross-sections (A = apical side, B = basal side). White arrowheads indicate IBA-1-positive cells located inside or at the choroidal (basal) side of the mutant RPE. Nuclei were counterstained with Hoechst. (**B**) Retinal sections from 6-months-old mice were stained for IBA-1 (red): increased signal intensity and presence of sub-retinal macrophages/microglia was detected in RPE$^{\Delta Abca1;Abcg1}$ mice (higher magnification images of the outer retina are shown in right panels). Nuclei were counterstained with DAPI. Representative images of N = 3 animals per group. *Il1b* (**C**), *Casp1* (**D**) and *Gfap* (**E**) mRNA levels were measured by semi-quantitative real-time PCR in neural retinas from Ctr and RPE$^{\Delta Abca1;Abcg1}$ mice. Shown are data from individual samples and means ± SD (N = 3–4). Statistics: one-way ANOVA vs '2 months' of the respective genotype; *: p<0.05, ****: p<0.0001. Abbreviations as in *Figure 1*.

DOI: https://doi.org/10.7554/eLife.45100.018

of IBA-1 positive cells across the RPE layer. Pigmentation of these cells could be due to phagocytosis of melanin granules-rich debris of RPE cells. Increased expression of interleukin 1β (*Il1b*), caspase 1 (*Casp1*) and glial fibrillary acidic protein (*Gfap*) in neural retinas of RPE$^{\Delta Abca1;Abcg1}$ mice confirmed a time-dependent inflammatory/stress response upon deletion of *Abca1* and *Abcg1* in the RPE (*Figure 9C–E*).

### Single *Abca1*, but not *Abcg1*, KO is sufficient to cause early lipid accumulation in the RPE

We initially generated double *Abca1;Abcg1* KO mice in order to completely block the active cholesterol efflux pathway in the RPE. To investigate the individual contribution of each gene to the phenotype, we generated RPE-specific *Abca1* (RPE$^{\Delta Abca1}$) and *Abcg1* (RPE$^{\Delta Abcg1}$) single mutant mice (*Table 1*). Analysis at 2 months of age showed that the RPE morphology of single RPE$^{\Delta Abca1}$ mice was similar to the double RPE$^{\Delta Abca1;Abcg1}$ mutants (*Figure 10A*). On the other hand, single RPE$^{\Delta Abcg1}$ mice were undistinguishable from the Ctr animals (*Figure 10A*). Furthermore, ORO staining confirmed accumulation of neutral lipids in RPE$^{\Delta Abca1}$ but not in RPE$^{\Delta Abcg1}$ mice (*Figure 10B*), even though CRE was similarly expressed in the RPE layer of all mutant mice (*Figure 10C*). Thus *Abca1* was the main driver of early morphological alterations and lipid accumulation in the RPE.

### Decreased *ABCA1* expression in human-derived cells carrying the AMD risk-conferring allele of *ABCA1*

Two SNPs in intron 2 of the human *ABCA1* gene (*rs1883025* and *rs2740488*), which are in high linkage disequilibrium (r$^2$ = 0.941), have been associated with AMD (*Chen et al., 2010*; *Fauser et al., 2011*; *Peter et al., 2011*; *Yu et al., 2011*; *Fritsche et al., 2016*). The major 'C' allele of rs1883025 and 'A' allele of rs2740488 have been described to confer increased risk for AMD, while the minor 'T' allele of rs1883025 and 'C' allele of 2740488 were associated with a decreased risk of AMD. However, the effect of these SNPs on ABCA1 expression and/or function remains unknown. To study the potential effect of the AMD-associated SNPs on *ABCA1* expression, we generated lymphoblastoid

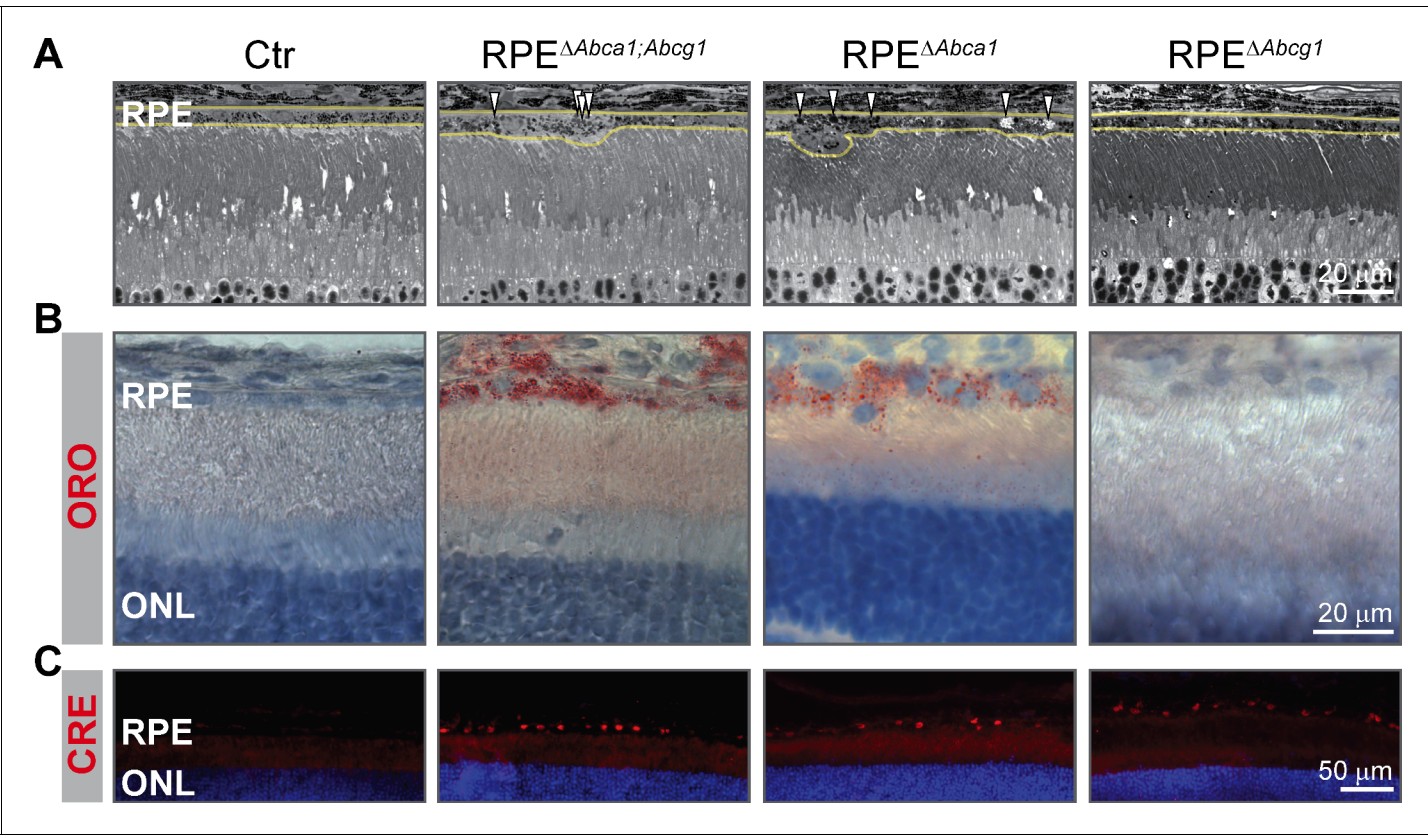

**Figure 10.** Early lipid accumulation in the RPE of single *Abca1*, but not *Abcg1*, KO mice. 2-months-old Ctr, double KO and single KO retinal sections were analyzed by light microscopy (A), ORO staining (B) and CRE IF (C). Single *Abca1* mutant mice (RPE$^{\Delta Abca1}$) showed an RPE phenotype comparable to double mutants (RPE$^{\Delta Abca1;Abcg1}$), while single *Abcg1* KO mice (RPE$^{\Delta Abcg1}$) were undistinguishable from controls. Yellow lines in (A) indicate RPE borders. Nuclei were counterstained with hematoxylin (B) or DAPI (C). Representative pictures of N ≥ 3 animals per group. Abbreviations as in *Figure 1*.

DOI: https://doi.org/10.7554/eLife.45100.019

cell lines (LCLs) from healthy individuals carrying homozygous decreased (N = 3) and increased risk (N = 3) genotypes for the SNPs (*Table 2*). *ABCA1* expression in LCLs was induced by LXR agonist stimulation and mRNA and protein levels were compared between LCLs carrying the different alleles. LCLs derived from subjects homozygous for the AMD increased risk allele of *ABCA1* showed significantly decreased *ABCA1* mRNA expression compared to reduced risk carriers (*Figure 11A*). A trend towards decreased ABCA1 expression was observed also at the protein level in carriers of the AMD increased risk genotype (*Figure 11B and C*). Even though the difference did not reach significance (p=0.14), probably due to the low sample numbers and intrinsic variability, these data provide the first indication of a potential correlation between AMD risk-associated genotypes and decreased *ABCA1* expression, which may impair cholesterol efflux from RPE cells in patients. This finding might be significant for a potential therapy aiming at *ABCA1* gene augmentation (see discussion).

## Discussion

Given the link between lipid metabolism and AMD, we generated and characterized a novel RPE-specific *Abca1;Abcg1* KO mouse model (RPE$^{\Delta Abca1;Abcg1}$). Although inactivation of the two genes was patchy due to variable *Cre* expression, genetic ablation of *Abca1* and *Abcg1* resulted in strong lipid accumulation in RPE cells (*Figures 2* and *5*). This is in agreement with the known function of ABCA1 and ABCG1 in mediating lipid efflux (*Cavelier et al., 2006*). Lipid accumulation was accompanied by morphological alterations and, at older ages, loss of RPE cells. Increased size and irregular shape of RPE cells in mutant mice (*Figure 3*) suggested that the healthy cells expanded in order to fill gaps in the epithelium that were generated by the drop out of CRE-positive cells and keep an intact barrier between neural retina and choroid, as previously described (*Nagai and Kalnins, 1996*; *Jiang et al., 2014*). Nevertheless, discontinuities in the RPE were observed in 6-months-old RPE-$^{\Delta Abca1;Abcg1}$ mice, together with degeneration of photoreceptors in the affected areas (*Figures 3* and *7*). We hypothesize that these were areas where numerous RPE cells were affected by CRE-mediated *Abca1;Abcg1* deletion, resulting in cell death and, therefore, in gaps too large to be filled by expanding neighboring cells. *Abca1* was the main responsible gene for maintaining lipid homeostasis and survival of RPE cells at 2 months of age, since lack of *Abca1*, but not *Abcg1*, was sufficient to cause strong lipid accumulation (*Figure 10*). This is in marked contrast to macrophages where both *Abca1* and *Abcg1* needed to be inactivated to observe a phenotype in non-stressed retinas (*Ban et al., 2018a*). *Abcg1* may thus be capable to compensate for the loss of *Abca1* in macrophages but may only have a limited ability to do so in the RPE. The reason for this is still unclear but a potential difference in transport substrate specificity between the two cell types can be postulated. Thus, additional experiments are required to conclusively dissect the individual contribution of the two genes to lipid accumulation and impairment of RPE function. Importantly, photoreceptor- and macrophage-specific *Abca1* and/or *Abcg1* KO mice showed a weaker retinal phenotype compared to RPE$^{\Delta Abca1;Abcg1}$ (*Sene et al., 2013*; *Ban et al., 2018a*; *Ban et al., 2018b*), suggesting that the lipid efflux pathway regulated by *Abca1* and *Abcg1* is of particular importance for the RPE. Furthermore, RPE cells may not be able to easily compensate for the absence of *Abca1* and *Abcg1* by activating alternative mechanisms. RNAseq data for example revealed only very minor alterations in the RPE- and retina-specific transcriptomes of 2-months-old RPE$^{\Delta Abca1;Abcg1}$ mice (data not shown). This suggests that absence of ABCA1 and ABCG1 in the RPE did not cause strong secondary gene expression changes that could balance the impaired lipid efflux pathway in RPE$^{\Delta Abca1;Abcg1}$ mice. Taken together, our data demonstrate that proper lipid handling by the RPE through active cholesterol efflux is essential for maintenance of an intact and functional retina *in vivo*. Moreover, local

**Table 2.** LCLs and genotypes of the AMD-associated SNPs in human *ABCA1* intron 2.

| LCL | SNP | Genotype |
| --- | --- | --- |
| *Decreased risk* (n = 3) | rs1883025 | TT |
| | rs2740488 | CC |
| *Increased risk* (n = 3) | rs1883025 | CC |
| | rs2740488 | AA |

DOI: https://doi.org/10.7554/eLife.45100.021

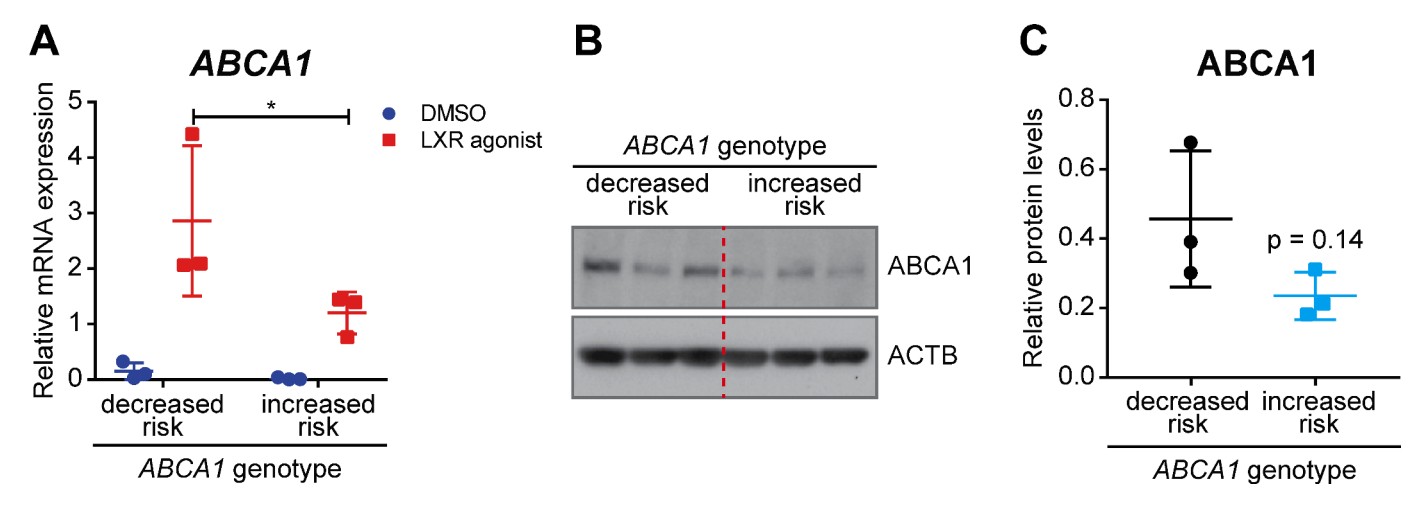

**Figure 11.** *ABCA1* expression in human LCLs. LCLs derived from healthy individuals carrying the AMD decreased or increased risk *ABCA1* genotypes were stimulated with an LXR agonist (1 μM) or DMSO vehicle control for 24 hr. (**A**) *ABCA1* mRNA levels were measured by semi-quantitative real-time PCR. Shown are data from individual samples and means ± SD (N = 3, three technical replicates per cell line). Statistics: two-way ANOVA with Sidak's multiple comparison test; *: p<0.05. ABCA1 protein levels were measured in LXR-stimulated cells by WB and normalized on ACTB levels. Shown are a representative WB (**B**) and the means ± SD of the band intensity quantification (N = 3, five technical replicates per cell line) (**C**). Statistics: Student's t-test vs 'decreased risk'.

DOI: https://doi.org/10.7554/eLife.45100.020

impairment of the ABCA1-mediated lipid transport activity in the RPE may provide the molecular basis for the genetic link of *ABCA1* to AMD and partially explain the contradictory association between systemic lipid levels and the disease (*van Leeuwen et al., 2018*).

As mentioned above, prominent intracellular accumulation of LDs was observed in RPE lacking ABCA1/ABCG1 (*Figure 2*). Biochemical analysis of these LDs showed specific accumulation of CEs and, to a lesser extent, REs, while UC as well as PLs, SLs and GLs remained unchanged (*Figure 5*). It is conceivable that the RPE continued to phagocytize lipid-rich OS also in the absence of functional ABCA1/ABCG1 to support photoreceptors. This hypothesis is supported by increased presence of fatty acids typical of OS membranes, such as docosahexaenoic acid (22:6), in the RPE of mutant mice (*Figure 5A and B*). The specific accumulation of esterified cholesterol, which is very important for retinal homeostasis (*Fliesler and Bretillon, 2010*; *Pikuleva and Curcio, 2014*), fits well with the lipid composition of human drusen and SDDs (*Haimovici et al., 2001*; *Wang et al., 2010*; *Spaide et al., 2018*) and with the high cholesterol content in rod OS (*Fliesler and Schroepfer, 1982*). The unchanged intracellular levels of UC in eyecups of 2-months-old mutant animals suggest that RPE cells esterified UC and fatty acids from OS disks to generate neutral CEs and REs that can be stored into LDs in an attempt to maintain intracellular UC levels below a toxic threshold (*Tabas, 2002*; *Lakkaraju et al., 2007*). Eventually, however, lipid concentration may become too high in the absence of a functional efflux pathway and lead to cell death. In contrast to the RPE, deletion of *Abca1* alone or in combination with *Abcg1* in hepatocytes, the main contributors to systemic lipid levels, not only affected plasma concentrations of UC and CEs, but also those of PLs, TGs and SLs (*Timmins et al., 2005*; *Chung et al., 2010*; *Iqbal et al., 2018*). This difference compared to the RPE suggests once more a cell-type dependent substrate specificity for the lipid efflux pathway or a remodeling of HDLs in the bloodstream, a process that does not occur within cells. Indeed, intracellular lipidomic changes in macrophages and endothelial cells that lacked *Abca1* and *Abcg1* were more similar to the changes identified in RPE cells of RPE$^{ΔAbca1;Abcg1}$ mice, including an accumulation of cholesterol, both in its un-esterified and esterified forms (*Westerterp et al., 2013*; *Westerterp et al., 2016*).

In addition to CEs, the abundance of REs was increased in our model, suggesting that lack of *Abca1* and *Abcg1* not only reduced lipid efflux but also affected intracellular handling of REs as intermediates of the visual cycle (*Kiser and Palczewski, 2016*). An increase in REs and fatty acids

may change the kinetics of the enzymes involved in the initial phases of the visual cycle (*Saari, 2012*). Moreover, altered RPE apical morphology (*Figure 2B*) could affect the physical interaction between RPE cells and photoreceptor OS, resulting in impaired internalization of incoming all-*trans* retinol intermediates. Once this step is achieved, however, the visual cycle seemed less affected as shown by similar amounts of regenerated RHO at later intervals after bleaching (*Figure 6*). Interestingly, AMD patients show delayed rod-mediated dark adaptation, suggesting visual cycle disturbance, already at early stages of the disease (*Owsley et al., 2001*; *Owsley et al., 2007*).

A 'cholesterol-recycling' mechanism involving transport of OS-derived cholesterol from the RPE back to the photoreceptors was proposed for the retina (*Tserentsoodol et al., 2006*). It is rather surprising that retinal function (*Figure 8*), localization of rod and cone markers (not shown), and lipid composition of the neural retina (*Figure 5C*) were not or not strongly affected in young RPE$^{\Delta Abca1;}$ $^{Abcg1}$ mice. Thus, photoreceptors seem capable to cope with an impaired lipid supply from RPE. Rods and cones could get enough cholesterol from the healthy CRE-negative RPE cells or they could re-direct towards a different lipid source like the intra-retinal circulation. Since retinal cells are able to synthesize cholesterol (*Fliesler and Bretillon, 2010*), functional cholesterol efflux from the RPE may not be absolutely required for photoreceptor survival. We therefore propose that photoreceptor loss in RPE$^{\Delta Abca1;Abcg1}$ mice is a secondary effect to dysfunctional RPE.

In summary, our model recapitulates some important features of dry AMD. i) Impaired lipid efflux in the RPE primarily affects RPE function and survival resulting in secondary photoreceptor degeneration and decreased retinal function in our mice. In both its dry and wet forms, AMD affects RPE cells while many photoreceptors in the macula may be lost secondarily (*Rattner and Nathans, 2006*; *Lim et al., 2012*). ii) The phenotype of RPE$^{\Delta Abca1;Abcg1}$ mice is age-related and slowly progressing, similar to AMD. iii) The photoreceptor/RPE layer of mutant mice at 4–6 months of age is infiltrated with inflammatory cells, an important hallmark of AMD pathology (*Kauppinen et al., 2016*).

In addition to the characterization of the mouse model, we present novel preliminary data on the effect of AMD risk-associated SNPs in *ABCA1* on its expression level. No variants in *ABCG1* have so far been associated with the disease, suggesting a predominant role of *ABCA1* in the RPE/retina, an interpretation that fits to the early phenotype of single KO mice in this study (*Figure 10*). Our data from human cells (*Figure 11*) suggest that the AMD increased risk allele correlates with lower *ABCA1* expression, at least upon LXR stimulation. The limited effect of the SNPs on *ABCA1* expression may not be surprising given their intronic location and the relatively small effect size of the SNPs on the disease (odds ratio 0.9 (*Fritsche et al., 2016*)). Variants in non-coding regions of the genome, including in the *ABCA1* locus (*Rhyne et al., 2009*), may directly change gene expression by affecting splicing, chromatin accessibility or binding of transcription factors (*Cooper, 2010*). On the other hand, we cannot exclude the possibility of an indirect effect due to regions inherited *in cis* with the SNPs or a difference between cell lines in their responsiveness to LXR stimulation. Clearly however, the potential effect of the SNPs on *ABCA1* expression should be confirmed in a larger study, ideally using RPE cells derived from induced pluripotent stem cells (iPSCs) (*Leach et al., 2016*; *Brandl, 2019*). Independently of the genotype, it has been reported that expression and function of *ABCA1* is reduced in aged mouse and human monocytes, including in the eye (*Sene et al., 2013*). Likewise, own preliminary data suggested a tendency of reduced expression of *ABCA1* in eyecups of old human donors (data not shown). It was also shown that cholesterol efflux was less efficient in old compared to young mouse RPE cells (*Biswas et al., 2017*), further suggesting an age-dependent physiological decline in *ABCA1* expression and function. In the presence of the risk-conferring *ABCA1* allele, expression of the gene may decrease below a critical threshold needed to prevent disease development. If so, this age-dependent decline could be targeted by the pharmacological activation of *ABCA1* gene expression, for example through treatment with an LXR agonist (*Koldamova et al., 2014*).

Besides the intronic variants being associated with AMD, biallelic mutations in the coding region of *ABCA1* are known to cause the very rare Tangier disease, a systemic condition characterized by virtual absence of plasma HDLs, cholesterol accumulation in several tissues and, in some instances, peripheral neuropathy and increased risk of developing cardiovascular disease (*Schaefer et al., 2016*). However and in contrast to our mouse data, Tangier patients are not known to have any ophthalmological phenotype, including AMD, except mild corneal opacities (*Winder et al., 1996*). RPE of Tangier patients might be healthier compared to AMD-affected RPE, making the impact of dysfunctional ABCA1 weaker in Tangier disease. This might be due to additional impaired mechanisms

present in aged/AMD RPE cells, such as oxidative stress, accumulation of bis-retinoids, genetic factors and others.

In conclusion, this study supports an essential role of the ABCA1/ABCG1 lipid efflux pathway for mouse RPE survival *in vivo* and suggests that an impaired lipid metabolism via ABCA1 may contribute to the pathology of AMD, most likely in combination with additional mechanisms. If the link between *ABCA1* and AMD is confirmed, activation of ABCA1-mediated lipid efflux will be an attractive target for AMD therapies.

# Materials and methods

## Key resources table

| Reagent type (species) or resource | Designation | Source or reference | Identifiers | Additional information |
|---|---|---|---|---|
| Gene (*Mus musculus*) | *Abca1* | | NCBI gene ID: 11303 | |
| Gene (*Mus musculus*) | *Abcg1* | | NCBI gene ID: 11307 | |
| Strain, strain background (*Mus musculus*) | C57BL/6J (wt) | The Jackson Laboratory | RRID: IMSR_JAX:000664; The Jackson Laboratory: 000664 | |
| Strain, strain background (*Mus musculus*) | BEST1Cre | *Iacovelli et al., 2011* | RRID:IMSR_JAX:017557 | Name at the Jackson Laboratory: C57BL/6-Tg (BEST1-cre)1Jdun/J |
| Strain, strain background (*Mus musculus*) | *Abca1*<sup>flox/flox</sup>; *Abcg1*<sup>flox/flox</sup> | The Jackson Laboratory | RRID:IMSR_JAX:021067 | Name at the Jackson Laboratory: B6.Cg-Abca1 tm1Jp Abcg1tm1Tall/J |
| Antibody | anti-ABCA1 (rabbit polyclonal) | Novus Biologicals | RRID:AB_10000630; Novus Biologicals: NB400-105 | (1:250 for IF, 1:200 for WB) |
| Antibody | anti-ABCG1 (rabbit monoclonal) | Abcam | RRID:AB_867471; Abcam: ab52617 | (1:100) |
| Antibody | anti-EZR (mouse monoclonal) | Santa Cruz Biotechnology | RRID:AB_783303; Santa Cruz: sc-58758 | (1:500) |
| Antibody | anti-CRE (rabbit polyclonal) | Merck | RRID:AB_10806983; Merck: 69050–3 | (1:300) |
| Antibody | anti-IBA1 (rabbit polyclonal) | Wako Fujifilm | RRID:AB_839504; Wako Fujifilm: 019–19741 | (1:500) |
| Antibody | anti-ZO1 (rabbit polyclonal) | Thermo Fisher Scientific | RRID:AB_2533456; Thermo Fisher Scientific: 40–2200 | (1:100) |
| Antibody | anti-βcatenin (mouse monoclonal) | BD Biosciences | RRID:AB_397554; BD Biosciences: 610153 | (1:300) |
| Antibody | anti-POU4F1 (mouse monoclonal) | Merck | RRID:AB_94166; Merck: MAB1585 | (1:100) |
| Recombinant DNA reagent | pTR-BEST1-Cre-P2A-GFP (AAV vector plasmid) | This paper | | Constructed from AAV plasmid materials at the University of Florida, laboratory of S. Boye |
| Sequence-based reagent | Random Primers | Promega | Promega: C1181 | |
| Peptide, recombinant protein | Phalloidin-Alexa488 | Thermo Fisher Scientific | RRID:AB_2315147; Thermo Fisher Scientific: A12379 | (1:100) |
| Commercial assay or kit | LipidTOX Red Neutral Lipid Stain | Thermo Fisher Scientific | Thermo Fisher Scientific: H34476 | (1:200) |

*Continued on next page*

*Continued*

| Reagent type (species) or resource | Designation | Source or reference | Identifiers | Additional information |
|---|---|---|---|---|
| Commercial assay or kit | Protease Inhibitos Cocktail | Sigma-Aldrich | Sigma-Aldrich: P2417 | |
| Commercial assay or kit | PowerUp Syber Green Master Mix | Thermo Fisher Scientific | Thermo Fishe rScientific: A25742 | |
| Commercial assay or kit | NucleoSpin RNA isolation kit | Macherey-Nagel | Macherey-Nagel: 740949.250 | |
| Chemical compound, drug | OilRedO (ORO) | Sigma-Aldrich | Sigma-Aldrich: O9755-25G | |
| Chemical compound, drug | Oxalic Acid | Sigma-Aldrich | Sigma-Aldrich: 75688 | |
| Chemical compound, drug | LXR agonist | Roche, *Panday et al., 2006* | Roche: T0901317 | |
| Chemical compound, drug | SPLASH | Avanti Polar Lipids | Avanti Polar Lipids: 330707 | |
| Chemical compound, drug | d7-sphinganine (SPH d18:0) | Avanti Polar Lipids | Avanti Polar Lipids: 860658 | D-erythro-sphinganine-d7 |
| Chemical compound, drug | d7-sphingosine (SPH d18:1) | Avanti Polar Lipids | Avanti Polar Lipids: 860657 | D-erythro-sphingosine-d7 |
| Chemical compound, drug | Dihydroceramide (Cer d18:0/12:0) | Avanti Polar Lipids | Avanti Polar Lipids: 860635 | N-lauroyl-D-erythro-sphinganine |
| Chemical compound, drug | Ceramide (Cer d18:1/12:0) | Avanti Polar Lipids | Avanti Polar Lipids: 860512 | N-lauroyl-D-erythro-sphingosine |
| Chemical compound, drug | Glucosylceramide (GluCer d18:1/8:0) | Avanti Polar Lipids | Avanti Polar Lipids: 860540 | D-glucosyl-ß—1,1'-N-octanoyl-D-erythro-sphingosine |
| Chemical compound, drug | Sphingomyelin (SM d18:1/12:0) | Avanti Polar Lipids | Avanti Polar Lipids: 860583 | N-lauroyl-D-erythro-sphingosylphosphorylcholine |
| Chemical compound, drug | d7-sphingosine-1-phosphate (S1P d18:1) | Avanti Polar Lipids | Avanti Polar Lipids: 860659 | D-erythro-sphingosine-d7-1-phosphate |
| Chemical compound, drug | Methanol | Honeywell | Honeywell: 34860 Riedel-de Haen | |
| Chemical compound, drug | MTBE | Sigma-Aldrich | Sigma-Aldrich: 20256 | tert-Butyl methyl ether |
| Chemical compound, drug | Chloroform | Sigma-Aldrich | Sigma-Aldrich: 650498 | |
| Chemical compound, drug | Acetonitrile | Sigma-Aldrich | Sigma-Aldrich: 534851 | |
| Chemical compound, drug | Isopropanol | Sigma-Aldrich | Sigma-Aldrich: 59300 | |
| Software, algorithm | ImageJ Tissue Cell Geometry macro | Institute for Research in Biomedicine, Barcelona, Spain | | http://adm.irbbarcelona.org/image-j-fiji |
| Software, algorithm | Relative Quantification Software | Thermo Fisher Cloud | | https://www.thermofisher.com/uk/en/home/digital-science/thermo-fisher-connect/all-analysis-modules.html |
| Software, algorithm | GraphPad Prism, version 7 | GraphPad | RRID:SCR_002798 | |
| Software, algorithm | Tracefinder Clinical 4.1 | Thermo Fisher Scientific | | |

*Continued on next page*

*Continued*

| Reagent type (species) or resource | Designation | Source or reference | Identifiers | Additional information |
|---|---|---|---|---|
| Other | transcend TLX I eluting pump | Thermo Fisher Scientific | | |
| Other | Q-Exactive | Thermo Fisher Scientific | | |
| Other | Mini-PROTEAN Precast Gels, 4–15% polyacrylamide | BioRad | BioRad: 4561086DC | |
| Other | C30 Accucore LC column | Thermo Fisher Scientific | Thermo Fisher Scientific: 7826–152130 | 150 mm * 2.1 mm * 2.6 µm |

## Mice and genotyping

All animal experiments adhered to the ARVO Statement for the Use of Animals in Ophthalmic and Vision Research and the regulations of the Veterinary Authorities of Kanton Zurich, Switzerland (study approval reference numbers: ZH141/2016 and ZH216/2015). Mice were maintained as breeding colonies at the Laboratory Animal Services Center (LASC) of the University of Zurich in a 14 hr: 10 hr light-dark cycle with lights on at six am and lights off at eight pm. Mice had access to food and water *ad libitum*. Average light intensity at cage levels was 60–150 lux, depending on the position in the rack. C57BL/6J (Bl6) were used as wild type controls. *BEST1Cre* mice were described earlier (*Iacovelli et al., 2011*). *Abca1;Abcg1* double floxed mice (*Abca1^{flox/flox};Abcg1^{flox/flox}*) were purchased from The Jackson Laboratory (Bar Harbor, ME, USA). Founder mice were on a Bl6 background and were genotyped for absence of known spontaneous mutations leading to retinal degeneration (*rd1*, *rd8*, *rd10*, *Cpfl1* and *Gpr179*). Mice were crossed in order to generate double- and single-floxed *Cre*-positive mice and *Cre*-negative littermate controls. All breeding pairs were heterozygous for *BEST1Cre*. Primers listed in *Supplementary file 1C* were used to genotype the mice by conventional PCR using genomic DNA extracted from ear biopsies or eye tissues. Although ocular expression of the *BEST1Cre* transgene is restricted to post-natal RPE (*Iacovelli et al., 2011* and *Figure 1C*), *BEST1Cre* can be expressed in other cell types, such as melanocytes (*Sundermeier et al., 2017*) and Sertoli cells of the testis (*Masuda and Esumi, 2010*; *Milenkovic et al., 2015*). Probably due to ectopic expression of the transgene in germ-line cells, we occasionally observed systemic or mosaic heterozygous KO animals for *Abca1* and/or *Abcg1* (*Figure 1—figure supplement 1A*). We controlled for presence of the excised allele in ear biopsies to avoid generation of full KO animals and defined our mice as shown in *Table 1*.

Since a heterozygous flox/- genotype resulted in a 50% reduction of the *Abca1* and *Abcg1* transcripts in non *Cre*-expressing tissues (*Figure 1—figure supplement 1B*), we excluded the possibility that systemic lack of one functional *Abca1* and/or *Abcg1* allele had an impact on the observed phenotype. To this aim, eyes of *Cre*-negative heterozygous animals (*Abca1^{flox/-};Abcg1^{flox/flox}*, *Abca1^{flox/flox};Abcg1^{flox/-}*, or *Abca1^{flox/-};Abcg1^{flox/-}*) were analyzed up to 6 months of age. No difference to *Abca1^{flox/flox};Abcg1^{flox/flox}* controls were found (retinal morphology in *Figure 1—figure supplement 1C* and ERG data not shown). All *BEST1Cre*-negative mice were therefore used as control animals.

## AAV generation and injection

A *Cre*-expression cassette was fused to *GFP* via a porcine teschovirus 2A (P2A) sequence and cloned downstream of the RPE-specific human *BEST1* promoter into the *pTR* vector. *pTR-BEST1-Cre-P2A-GFP* was packaged into AAV4 capsid at the Viral Vector Facility of the Neuroscience Center Zurich (ZNZ), University of Zurich, Switzerland. $7.3 \times 10^9$ viral genomes/eye (1 µL volume) were injected into the sub-retinal space of *Abca1^{flox/flox};Abcg1^{flox/flox}* mice as previously described (*Barben et al., 2018a*). Mice were injected at 4–15 weeks of age and sacrificed 10 weeks post-injection. Eyes were marked nasally by cauterization and fixed for subsequent IF/lipid staining as described below.

## Morphology, light microscopy and transmission electron microscopy

Eyes were marked dorsally by cauterization and prepared as described (*Barben et al., 2018a*). 500 nm nasal-temporal sections were analyzed by light microscopy (Zeiss Axioplan, Feldbach,

Switzerland) and Adobe Photoshop CS6 (Adobe Systems Inc, San Jose, CA, USA) was used to pho-tomerge high magnification images of the outer and inner retina as well as to create retina panoramas. Images at higher magnification were always acquired from the central region close to the optic nerve head. The ruler tool of Adobe Photoshop CS6 was used to measure ONL and INL thickness at the indicated distance from optic nerve head in retinal panoramas. For transmission electron microscopy, ultrathin sections (50 nm) were cut, stained with uranyl acetate and lead citrate and analyzed using a Philips CM100 transmission electron microscope (Philips, Amsterdam, The Netherlands).

## IF on retinal sections, ORO staining and RPE flat mounts

Eyes were marked dorsally by cauterization and retinal 12 µm nasal-temporal cryosections were prepared as described (*Barben et al., 2018b*). For AAV-injected animals, eyes were marked nasally and dorsal-ventral sections were cut. Sections were blocked in blocking solution (3% normal goat serum, 0.3% Triton X-100 in 0.1 M phosphate buffer (PB)) for 1 hr at room temperature (RT), followed by overnight incubation at 4°C with the following primary antibodies: rabbit anti-ABCA1 (1:250, NB400-105, Novus Biologicals, Littleton, CO, USA), rabbit anti-ABCG1 (1:100, ab52617, Abcam, Cambridge, UK), mouse anti-EZR (1:500, sc-58758, Santa Cruz Biotechnology, Dallas, TX, USA), rabbit anti-CRE (1:300, 69050–3, Merck, Darmstadt, Germany), rabbit anti-IBA-1 (1:500, 019–19741, Wako Fujifilm, Neuss, Germany) or mouse anti-POU4F1 (1:100, MAB1585, Merck). After three washing steps in PB salt (0.1 M PB with the addition of 0.8% NaCl and 0.02% KCl), samples were incubated at RT for 2 hr with appropriate secondary antibodies conjugated to Cy2, Cy3 or AlexaFluor555 fluorophores (Jackson ImmunoResearch, Suffolk, UK and Thermo Fisher Scientific, Reinach, Switzerland). Nuclei were counterstained with 4′,6-Diamidine-2′-phenylindole di-hydrochloride (DAPI, Thermo Fisher Scientific), sections were mounted with Mowiol and imaged using a fluorescent microscope (Zeiss Axioplan). Sections stained with secondary antibody only were used as negative controls.

For neutral lipid ORO staining, cryosections were washed with distilled $H_2O$ and incubated in 0.2% $KMnO_4$ for 40 min at RT, followed by neutralization with fresh 1% oxalic acid for 1–2 min to bleach the melanin pigment in the RPE/choroid. After two washing steps in $H_2O$, sections were rinsed with 60% isopropanol and incubated for 10 min at RT in 0.42% ORO working solution (Sigma-Aldrich, Merck, Buchs SG, Switzerland; 0.7% ORO stock solution in isopropanol diluted 3:2 in $H_2O$ to generate the working solution). Sections were rinsed with 60% isopropanol, washed twice with $H_2O$ and nuclei were counterstained with enhanced Meyer's hematoxylin (Artechemis, Zofingen, Switzerland) for 1–2 min. Sections were mounted with Mowiol and imaged using light microscopy within 15 days (Leica Microsystems, Heerbrugg, Switzerland).

RPE flat mounts were prepared as described (*Oczos et al., 2014*). After washing, samples were incubated for 1 hr at RT in blocking solution (see above), followed by overnight incubation at 4°C with primary antibodies: rabbit anti-CRE (see above), rabbit anti-ZO-1 (1:100, 40–2200, Thermo Fisher Scientific), mouse anti-β-cat (1:300, 610153, BD Biosciences, Allschwil, Switzerland) or rabbit anti-IBA-1 (see above). After three washing steps in PB salt, samples were incubated at RT for 2 hr with appropriate secondary antibodies as described above or phalloidin-AlexaFluor488 to stain F-actin (1:100, A12379, Thermo Fisher Scientific). Nuclei were counterstained with Hoechst (2 µg/ml, Sigma-Aldrich) and lipids with LipidTOX (1:200, H34476, Thermo Fisher Scientific) for 30 min at RT. Samples were mounted on glass slides with Mowiol and imaged using a fluorescent microscope (Zeiss Axioplan) or an SP8 inverted confocal microscope (Leica Microsystems). Three ZO-1-stained images per RPE flat mount quadrant (dorsal, ventral, nasal and temporal of the optic nerve head) were used for quantification with the Tissue Cell Geometry macro in ImageJ (developed by the Institute for Research in Biomedicine, Barcelona, Spain, http://adm.irbbarcelona.org/image-j-fiji). At least N = 998 RPE cells per group (N = 3–4 mice) were examined. The ratio between the major and minor axis of the fitted ellipse was used as a readout of cell shape.

## Plasma and eye tissue collection for lipid analysis

After a lethal dose of anesthesia, blood was collected by cardiac puncture using a 1 ml syringe and 26G needle into Microtainer $K_2$-EDTA-coated tubes (BD Biosciences). Tubes were inverted 20 times, plasma was separated by centrifugation at 2'500 g for 10 min at RT and snap-frozen in liquid nitrogen ($N_2$). Neural retinas were isolated through a slit in the cornea and snap-frozen in liquid $N_2$; corresponding eyecups (containing RPE) were isolated and dissected from contaminating cornea, optic

nerve or adipose tissue left overs. For analysis of UC, CEs and REs, tissues from both eyes of the same animal were pooled; whereas for analysis of PLs, SLs and GLs, tissues from single eyes were analyzed. For analysis of UC, CEs and REs, eyecup samples were enriched for RPE cells by incubating the tissues in 100 µl of PBS for 20 min at RT followed by flicking of the tubes 50 times to release pigmented cells into the PBS, similar to a procedure previously used for protein isolation (*Wei et al., 2016*). Remaining posterior eyecups were removed and samples snap-frozen in liquid $N_2$. These samples were labelled as 'RPE-enriched eyecup' (*Figure 5*). For analysis of PLs, SLs and GLs, complete eyecups were snap-frozen in liquid $N_2$. These samples were labelled as 'whole eyecup' (*Figure 5*). After thawing, 100 µl of PBS were added to each tissue. All samples were then homogenized by sonication, 20 µl of 0.6% Triton in PBS were added to each tube (final concentration: 0.1% Triton) and samples were incubated on a rotating wheel for 1 hr at 4°C. Samples were centrifuged at 1'000 g for 3 min at RT and supernatant used for protein quantification using the bicinchoninic acid assay (BCA, Thermo Fisher Scientific) followed by lipid extraction.

## Lipid extraction

Lipid extraction was performed as described previously (*Pellegrino et al., 2014*) with some modifications. For UC, CEs and REs, 1 ml of a methanol:MTBE:chloroform (MMC) mixture 4:3:3 (v/v/v) was added to 20 µl plasma or 50 µg protein of tissue homogenate. The MMC mix was fortified with 100 pmoles of d7-cholesterol and d7-CE 16:0 (Avanti Lipids, Alabaster, AL, USA). Samples were briefly vortexed and mixed on a shaker at 37°C (1'400 rpm, 20 min). Protein precipitation was obtained after centrifugation for 5 min, 16'000 g, 25°C. The single-phase supernatant was collected, dried under $N_2$ and stored at −20°C until analysis. Dried lipids were dissolved in 100 µl methanol. For PLs, SLs and GLs, 1 ml of MMC mixture 1.33:1:1 was added to 20 µl of plasma or tissue homogenate. The MMC was fortified with the SPLASH mix of internal standards and 100 pmoles/ml of the following internal standards (all from Avanti Lipids): d7-sphinganine (SPH d18:0), d7-sphingosine (SPH d18:1), dihydroceramide (Cer d18:0/12:0), ceramide (Cer d18:1/12:0), glucosylceramide (GluCer d18:1/8:0), sphingomyelin (SM d18:1/12:0) and 50 pmoles/ml d7-sphingosine-1-phosphate (S1P d18:1). Samples were briefly vortexed and mixed on a shaker at 25°C (950 rpm, 30 min). Protein precipitation was obtained after centrifugation for 10 min, 16'000 g, 25°C. The single-phase supernatant was collected, dried under $N_2$ and stored at −20°C until analysis. Dried lipids were dissolved in 100 µL methanol:isoproanol (1:1, v/v).

## Lipid analysis

Liquid chromatography was done according to (*Narváez-Rivas and Zhang, 2016*) with some modifications. Lipids were separated using a C30 Accucore LC column (150 mm * 2.1 mm * 2.6 µm) and a transcend TLX eluting pump (Thermo Fisher Scientific). UC, CEs and REs were separated with the following mobile phases: A) acetonitrile:water (2:8 v/v) with 10 mM ammonium acetate and 0.1% formic acid, B) isopropanol:acetonitrile (9:1 v/v) with 10 mM ammonium acetate and 0.1% formic acid and C) methanol at a flow rate of 0.3 ml/min. The following gradient was applied: 0.0–1.5 min (isocratic 70% A, 20% B and 10% C), 1.5–18.5 min (ramp 20–100% B), 18.5–25.5 min (isocratic 100% B) and 25.5–30.5 min (isocratic 70% A, 20% B and 10% C). PLs, SLs and GLs were separated with the following mobile phases: A) acetonitrile:water (6:4 v/v) with 10 mM ammonium acetate and 0.1% formic acid and B) as above at a flow rate of 0.26 ml/min. The following gradient was applied: 0.0–0.5 min (isocratic 30% B), 0.5–2 min (ramp 30–43% B), 10–12.0 min (ramp 43–55% B), 12.0–18.0 min (ramp 65–85% B), 18.0–20.0 min (ramp 85–100% B), 20–35 min (isocratic 100% B), 35–35.5 min (ramp 100–30% B) and 35.5–40 min (isocratic 30% B).

The liquid chromatography was coupled to a hybrid quadrupole-orbitrap mass spectrometer Q-Exactive (Thermo Fisher Scientific). For UC, CEs and REs, samples were analyzed in positive mode using a heated electrospray ionization (HESI) interface. The following parameters were used: spray voltage 3.5 kV, vaporizer temperature of 300°C, sheath gas pressure 20 AU, aux gas 8 AU and capillary temperature of 320°C. The detector was set to an MS2 method using a data-dependent acquisition with top10 approach with stepped collision energy between 25 and 30. A 140'000 resolution was used for the full spectrum and a 17'500 for MS2. A dynamic exclusion filter was applied which excluded fragmentation of the same ions for 20 s. For PLs, SLs and GLs, a data-dependent acquisition with positive and negative polarity switching was used. A full scan was used from 220 to 3'000

m/z at a resolution of 70'000 and AGC Target 3e6 while data-dependent scans (top10) were acquired using normalized collision energies (NCE) of 25, 30 and a resolution of 17'500 and AGC target of 1e5.

Identification criteria for UC, CEs and REs were 1) resolution with an accuracy of 5 ppm from the predicted mass at a resolving power of 140'000 at 200 m/z, 2) matching retention time on synthetic available standards and 3) the specific fragmentation patterns ([M-H2O]+ and 369.3 for cholesterol esters and 269.2 for retinyl esters). Identification criteria for PLs, SLs and GLs were 1) resolution with an accuracy of 5 ppm from the predicted mass at a resolving power of 70'000 at 200 m/z, 2) isotopic pattern fitting to expected isotopic distribution, 3) comparison of the expected retention time to an in-house database and 4) fragmentation pattern matching to an in-house experimentally validated lipid fragmentation database. Quantification was done using single point calibration or by comparing the area under the peak of each species to the area under the peak of the internal standard. Quality controls using a mixture of all samples were used in four concentration (1x, 0.5x, 0.25x and 0.125x). Triplicates on the quality controls were measured, and the CV% for each of the lipids reported was below 20%. Mass spectrometric data analysis was performed in Treacefinder software 4.1 (Thermo Fisher Scientific) for peak picking, annotation and matching to the in-house fragmentation database.

## Fundus imaging/OCT and ERG

Pupils were dilated using Cyclogyl 1% (Alcon Pharmaceuticals, Fribourg, Switzerland) and Neosynephrine 5% (Ursapharm Schweiz GmbH, Roggwil, Switzerland) 20 min prior to anesthesia. Mice were anesthetized by subcutaneous injection of ketamine (85 mg/kg, Parke-Davis, Berlin, Germany) and Xylazine (4 mg/kg, Bayer AG, Leverkusen, Germany) and a drop of 2% Methocel (OmniVision AG, Neuhausen, Switzerland) was applied to keep the eyes moist. Mice were placed on a heated pad and fundus images and OCT scans were acquired using the Micron IV system (Phoenix Research Labs, Pleasanton, CA, USA).

ERG recordings were performed as described (Kast et al., 2016). Briefly, mice were dark-adapted overnight, pupils dilated and animals anesthetized as described above. A drop of Mydriaticum Dispersa (OmniVision AG) was applied to induce mydriasis and to keep the tissue moist. A reference electrode was inserted subcutaneously between the eyes, a ground electrode was inserted subcutaneously at tail base and recording gold electrodes were placed onto mouse corneas. Mice were placed on a heated pad in front of a Ganzfeld chamber. Responses to 14 different light intensities ranging from $-50$ db (0.000025 cd*s/m$^2$) to 15 db (79 cd*s/m$^2$) for scotopic and eight different light intensities ranging from $-10$ db (25 cd*s/m$^2$) to 25 db (790 cd*s/m$^2$) for photopic conditions were recorded using an LKC UTAS Bigshot recording unit (LKC Technologies Inc, Gaithersburg, MD, USA). Mice were light-adapted for 5 min before photopic recordings. Ten recordings were averaged per light intensity; responses from the left and right eye of the same animal were averaged for subsequent analysis.

## Measurement of rhodopsin regeneration kinetics

All mice used for this experiment were homozygous for the $Rpe65_{450Met}$ variant. RHO regeneration was measured as previously described (Wenzel et al., 2005; Samardzija et al., 2008). Briefly, mice were dark-adapted overnight. After pupil dilation, mice were exposed to 5'000 lux of white light for 10 min, a light intensity and exposure duration that does not induce retinal damage in these mice. Mice were returned to darkness for the indicated time points (30, 60 or 120 min) or euthanized immediately. After euthanasia, retinas were isolated in darkness through a slit in the cornea and snap-frozen in N$_2$. RHO content was measured as described (Wenzel et al., 2005).

## Human subject recruitment, LCL generation and culture

The study was approved by the local ethical committee at the Radboud University Medical Center and was performed in accordance with the tenets of the Declaration of Helsinki. Individuals were selected from the European Genetic Database (EUGENDA, https://www.eugenda.org/), a large multicenter database for clinical and molecular analysis of AMD, and provided written informed consent before participation. Disease status was determined based on classification of color fundus photographs and, if available, spectral domain OCT and fluorescein angiography by certified graders as

previously described (*Ristau et al., 2014*). LCLs were generated for six control subjects, defined as individuals having only pigmentary changes, less than 10 small drusen or without macular abnormalities. Human B-lymphocytes were immortalized by transformation with the Epstein-Barr virus according to established procedures (*Wall et al., 1995*). LCLs were generated for three control individuals who were homozygous for *ABCA1* genotypes conferring decreased risk for AMD (*rs1883025* TT and *rs2740488* CC) and for three control individuals who were homozygous for *ABCA1* genotypes conferring increased risk for AMD (*rs1883025* CC and *rs2740488* AA), as shown in *Table 2*. DNA samples were genotyped with a custom-modified Illumina HumanCoreExome array (Illumina, Eindhoven, Netherlands) at the Center for Inherited Disease Research (CIDR, Baltimore, MD, USA) and quality control and genotype imputation using the 1000 Genomes Project reference panel (*Abecasis et al., 2012*) were performed by the International AMD Genomic Consortium as previously described (*Fritsche et al., 2016*). *ABCA1 rs1883025* genotypes were additionally confirmed by sequencing of a PCR fragment flanking the SNP on genomic DNA extracted from $1 \times 10^6$ cells. Sequencing results matched the human *ABCA1* locus, proving the human origin of the cell lines (data not shown). LCLs were also checked for absence of mycoplasma contamination via PCR using primers specific for the mycoplasma genome in the medium of confluent cultures (data not shown).

LCLs were cultured in a humid incubator at 37°C and 5% $CO_2$ in RPMI 1640 medium (Sigma-Aldrich) supplemented with 15% heat-inactivated fetal bovine serum (Gibco, Thermo Fisher Scientific), 20 mM HEPES buffer (Sigma-Aldrich) and 10'000 U/mL penicillin-streptomycin (Gibco). Cells were seeded at a concentration of $0.5-1 \times 10^6$ cells/ml and split every 3–4 days. For experiments, $2-3 \times 10^6$ cells per condition were seeded in 6-well plates and stimulated with 1 μM LXR agonist (T0901317; prepared at Roche as previously reported (*Panday et al., 2006*)) or DMSO vehicle control for 24 hr. LCLs were then washed with PBS and harvested for RNA or protein analysis (see below).

## Gene expression analysis

RNA was extracted from neural retina and eyecups (containing RPE and choroid) using an RNA isolation kit (Macherey-Nagel, Oensingen, Switzerland) with on column DNaseI treatment and used for cDNA synthesis with oligo-dT as previously described (*Samardzija et al., 2006*; *Storti et al., 2017*). For human LCL samples, RNA was isolated as above but 0.5 μg random primers (Promega, Dübendorf, Switzerland) were used instead of oligo-dT for cDNA synthesis. Transcript levels in 10 ng of cDNA were measured by semi-quantitative real-time PCR using an ABI QuantStudio3 machine (Thermo Fisher Scientific) with the PowerUp Sybr Green master mix (Thermo Fisher Scientific) and primer pairs specific for the genes of interest (*Supplementary file 1D*). Primers were designed to span large introns and avoid known SNPs. Beta-actin (*Actb*) was used to normalize mouse gene expression with the comparative threshold cycle method ($\Delta\Delta C_t$) of the Relative Quantification software of the Thermo Fisher Cloud. For LCL samples, *ACTB* and *RPL28* levels were used for double normalization with the same method. Note that in order to measure possible decrease in *Abca1* and *Abcg1* transcripts in KO mice, primers were designed to amplify part of the excised region (exons 45 and 46 for *Abca1* and exon 3 for *Abcg1*).

## Protein isolation from LCLs and Western Blotting (WB)

Cells were collected, washed twice with ice-cold PBS and lysed in 50 μl of RIPA buffer supplemented with protease inhibitor cocktail (P2417, Sigma-Aldrich) for 15 min on ice. After centrifugation at 16'000 g for 15 min at 4°C, supernatant was collected and protein concentration measured by BCA. 50 μg of proteins were loaded on 4–15% polyacrylamide gradient gels (Bio-Rad, Cressier, Switzerland) for SDS-PAGE followed by semi-dry transfer to a nitrocellulose membrane. Membranes were blocked in 5% non-fat blocking milk (Bio-Rad) for 1 hr at room temperature prior to incubation overnight at 4°C with primary antibodies: rabbit anti-ABCA1 (1:200, NB400-105, Novus Biologicals) and mouse anti-ACTB (1:10'000, A5441, Sigma-Aldrich). After washing, membranes were incubated with appropriate horseradish peroxidase (HRP)-conjugated secondary antibodies for 1–2 hr at RT. Signals were developed using enhanced chemiluminescence (ECL) substrate (PerkinElmer, Schwerzenbach, Switzerland) and visualized using X-ray films. Intensity of bands was quantified using ImageJ and normalized to ACTB levels.

## Statistical analysis

The number of biological replicates is defined in figure legends as 'N' and refers to the number of individual animals or cell lines analyzed in this study. The number of technical replicates may be indicated in the corresponding figure legend as well, when appropriate. All statistical analysis, as indicated in figure legends, were performed using GraphPad Prism 7 (San Diego, CA, USA).

# Acknowledgments

The authors would like to thank Cornelia Imsand, Sarah Nötzli, Adrian Urwyler, Andrea Gubler and Ana Bordonhos (University of Zurich) for excellent technical support and Dr. Everson Nogoceke (Roche pRED) for helpful discussions.

# Additional information

## Competing interests

Christoph Ullmer: Employee of F Hoffmann-La Roche Ltd. Jürgen Fingerle, Cyrille Maugeais: Previous employee of F Hoffmann-La Roche Ltd. The other authors declare that no competing interests exist.

## Funding

| Funder | Grant reference number | Author |
| --- | --- | --- |
| Vontobel-Stiftung | | Federica Storti |
| Roche | RPF 378 | Federica Storti<br>Christian Grimm |
| Schweizerischer Nationalfonds zur Förderung der Wissenschaftlichen Forschung | 31003A_173008 | Vyara Todorova<br>Marijana Samardzija<br>Maya Barben<br>Christian Grimm |

The funders had no role in study design, data collection and interpretation, or the decision to submit the work for publication.

## Author contributions

Federica Storti, Conceptualization, Formal analysis, Validation, Investigation, Visualization, Methodology, Writing—original draft, Project administration; Katrin Klee, Vyara Todorova, Investigation, Methodology, Writing—review and editing; Regula Steiner, Alaa Othman, Resources, Formal analysis, Investigation, Methodology, Writing—review and editing; Saskia van der Velde-Visser, Resources, Collection of human lymphocytes and generation of LCLs; Marijana Samardzija, Conceptualization, Supervision, Investigation, Writing—review and editing; Isabelle Meneau, Frank Blaser, Resources, Collection of human donor tissues; Maya Barben, Duygu Karademir, Valda Pauzuolyte, Investigation, Methodology; Sanford L Boye, Resources, Generation of the pTR-BEST1-Cre-P2A-GFP (AAV vector plasmid); Christoph Ullmer, Supervision, Funding acquisition, Writing—review and editing; Joshua L Dunaief, Conceptualization, Resources, Writing—review and editing; Thorsten Hornemann, Conceptualization, Resources, Methodology, Writing—review and editing; Lucia Rohrer, Arnold von Eckardstein, Conceptualization, Resources, Funding acquisition, Writing—review and editing; Anneke den Hollander, Conceptualization, Resources, Investigation, Writing—review and editing; Jürgen Fingerle, Conceptualization, Supervision, Funding acquisition, Writing—review and editing; Cyrille Maugeais, Conceptualization, Resources, Supervision, Funding acquisition, Project administration, Writing—review and editing; Christian Grimm, Conceptualization, Resources, Formal analysis, Supervision, Funding acquisition, Writing—original draft, Project administration, Writing—review and editing

## Author ORCIDs
Federica Storti (iD) http://orcid.org/0000-0002-5764-1656
Sanford L Boye (iD) http://orcid.org/0000-0002-8803-9369
Christian Grimm (iD) http://orcid.org/0000-0001-9318-4352

## Ethics
Human subjects: The study was approved by the local ethical committee at the Radboud University Medical Center, The Netherlands, and was performed in accordance with the tenets of the Declaration of Helsinki. Individuals were selected from the European Genetic Database (EUGENDA, https://www.eugenda.org/), a large multicenter database for clinical and molecular analysis of AMD, and provided written informed consent before participation.

Animal experimentation: All animal experiments adhered to the ARVO Statement for the Use of Animals in Ophthalmic and Vision Research and the regulations of the Veterinary Authorities of Kanton Zurich, Switzerland (study approval reference numbers: ZH141/2016 and ZH216/2015).

## Decision letter and Author response
Decision letter https://doi.org/10.7554/eLife.45100.025
Author response https://doi.org/10.7554/eLife.45100.026

# Additional files
## Supplementary files
• Supplementary file 1. Supplementary tables. (**A**) Absolute concentrations of analyzed lipid classes from 2-months-old Ctr and RPE$^{\Delta Abca1;Abcg1}$ mice. (**B**) RHO measurements from 2-months-old Ctr and RPE$^{\Delta Abca1;Abcg1}$ neural retinas. (**C**) Primers used for genotyping. (**D**) Primers used for gene expression analysis.
DOI: https://doi.org/10.7554/eLife.45100.022

• Transparent reporting form
DOI: https://doi.org/10.7554/eLife.45100.023

## Data availability
All data generated or analysed during this study are included in the manuscript and supporting files.

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
