## [Decision Letter]

Thank you for submitting your manuscript "Impaired ABCA1/ABCG1-mediated lipid efflux in the mouse retinal pigment epithelium (RPE) leads to retinal degeneration" to *eLife*. Three experts reviewed your manuscript, and their assessments, together with my own, form the basis of this letter. As you will see, all of the reviewers were impressed with the importance and novelty of your work.

The major item in our assessment is that we think that the post-mortem human data in Supplementary figure S5 are not compelling. One can see the issue by considering what the best fitting line would look like if the data point from the individual who died at ~age 30 is removed – the best fitting line would then be nearly horizontal. Our strong recommendation is to remove these data from the manuscript. A second recommendation is to immunostain WT vs KO mouse RPE for the transporters to assess the KO at the protein level. In the manuscript, the analysis of Cre expression is nice but it is one step removed from the transporter. Also, the PCR analysis of the recombined fragment does not assess the fraction of cells in which recombination did not occur. A third recommendation – and this one is less critical than the first two – is to conduct RNAseq on the WT vs KO RPE to see if there are secondary gene expression changes in response to transporter KO. Such changes, if they exist, could by informative.

I am including the three reviews at the end of this letter, as there are a variety of specific and useful suggestions in them. We appreciate that the reviewers' comments cover a range of suggestions for improving the manuscript. We look forward to receiving your revised manuscript.

Reviewer #1

This study demonstrates the importance of the ABCA1-mediated lipid efflux pathway in maintaining RPE and photoreceptor cell health. Specifically, they show unequivocally that lack of ABCA1 in RPE cells resulted in lipid droplet accumulation within the RPE which, over time, led to loss of the RPE cells. The authors generated and characterized an RPE-specific Abca1;Abcg1 double knockout mice on a *BEST1Cre* background, they used 2 month old mice to show evidence of Cre mRNA expression in the eyecup and very little in the retina, IF positive staining for Cre only in the RPE nuclei, and showed CRE-mediated excision fragments of Abca1 and Abcg1 in the eyecups with some variability, likely due to the variable Cre expression in the RPE of these mice. Also, at 2 months of age, on a normal diet, morphologically there were already changes: a dotted pattern on fundus images, changes seen in the apical border of the RPE and lipid droplets in the RPE, lipidTOX and OilRedO staining coinciding with Cre expression in the RPE. A decrease in the rates of rhodopsin regeneration after bleaching demonstrated that this was slower initially in the double knockout mice suggesting impairment of visual cycle even in these 2 month old mice.

Age-dependent changes were also seen by studying the mice at 2, 4 and 6 months. They detected deteriorating phenotypic changes by careful measurements of RPE shape, size, and Mct3 expression, light microscopy and EM showed increasing damage to the RPE and neural retina and increasing visual deficits by ERG measurements of 2, 4, and 6 month old mice.

They used the correct controls to ensure changes seen in the mice were not developmental, they injected an AAV virus expressing CRE and GFP into adult Abca1flox/flox:Abcg1flox/flox mice and they saw, in regions that were GFP-positive, the same changes in accumulation of lipids and RPE damage they saw with the double knockout mice. To ensure the phenotype of the double KO was not due to CRE toxicity the *BEST1CRE* mouse was used as a control. They also generated RPE-specific Abca1 KO and RPE-specific Abcg1 KO mice and studying these at 2 months of age determined that the Abca1 appeared to be the dominant gene responsible for lipid efflux in the RPE cells, resulting in lipid accumulation in the cells, this may imply a somewhat different regulation of lipid homeostasis in RPE cells compared to macrophages that require both Abca1 and Abcg1.

Overall a careful study which does not overstate the findings but nicely points to the ABCA1-mediated efflux of lipids as being essential for maintaining RPE health.

It is unclear which of the analyses shown in Figure 5 were from "RPE-enriched samples", i.e. from an eyecup without the neural retina, to which PBS was added and then the tubes were flicked to release RPE cells and which analysis used the entire eyecup without the neural retina. Were all the analyses done on single eyes or were the tissue lysates pooled?

In Figure 9A using 4 month old mice and flat mounts, sub-RPE IBA-1 positive cells infiltrating the RPE from the choroidal side are shown in flatmounts, can you give an indication of how many cells were seen and if you saw this in all the KO mice at this age. Then IBA-1 cells in 6 month mice showed, in retinal sections, sub-retinal immune cells were seen- were these seen in parts of the retina that still had an intact RPE layer or, as was seen in EM of sections of mice at this age had the RPE completely atrophied and disappeared. In the text, it appears there is a comparison made between the localization of immune cells at 4 months and 6 months, but because the figures show flat mounts for one time point and cross sections for the other the comparison is a little difficult to interpret.

In Supplementary figure S5 using eyecup tissue from 25 human donor eyes, the authors attempted to show that there was a decrease in ABCA1 mRNA expression levels with age. Only 4 of these samples were from donors younger than 53 and without these 4 samples there did not appear to be any decline in ABCA1 expression from 57 to 96 years of age. This was the least convincing part of this project and slightly detracted from the rest of the work.

Introduction: Change 'choroids' to choroid

Discussion section: Replace 'phagocyte' with 'phagocytize'

Discussion section: Replace 'synthetize' with 'synthesize'

Discussion section: Change 'In presence' to 'In the presence'

Reviewer #2

The manuscript by Storti et al., includes the characterization of a conditional knockout mouse that lacks ABCA1 and its effector protein ABCG1 in the RPE cells. RPEΔAbca1;Abcg1 mice presented an early onset ocular phenotype including lipid deposition in the RPE, local inflammatory reactivity, and progressive loss of both RPE and photoreceptor cells. Lymphoblastoid cells lines (LCL) from individuals with known AMD-risk variant of ABCA1 showed reduced ABCA1 expression when stimulated with a liver x receptor (LXR) agonist, further suggesting the role of ABCA1 in lipid metabolism. The manuscript is well-written for most of the parts. The study is more descriptive rather than providing new mechanistic insights on AMD. Given the similarities to AMD in the phenotypic features, the RPEDAbca1;Abcg1 mouse may be useful to test various therapeutic approaches targeting intracellular lipid/cholesterol metabolism. However, a major flaw of the study is the lack of direct investigations of ABCA1 and ABCG1 protein levels in the RPEΔAbca1;Abcg1 mouse by immunoblotting and/or immunohistochemistry (IHC).

The ABCA1 and ABCG1 protein levels were evaluated by IHC only in the wild-type sections (Figure 1A). For the RPEΔAbca1;Abcg1 mouse, the read out for gene knockdown was evidenced primarily by Cre mRNA expression (Figure 1B and Figure 3F; Figure 1—figure supplement 2 and Figure 3—figure supplement 3) and at the protein level by IHC using an antibody against Cre rather than ABCA1 and ABCG1. Expression of Abca1 and Abcg1 mRNA (Figure 1—figure supplement 1 and Figure 1—figure supplement 2) was reduced in the mutated mouse but no protein data is shown.

Cre-driven morphological changes and increased lipid deposition in the RPE cells RPEΔAbca1;Abcg1 mouse were clearly evidenced in Figure 2 and Figure 5. The group also choose an AAV-based approach to transduce the RPE cells of Abca1 and Abcg1 Floxed/Floxed mouse to further draw a correlate between the GFP-Cre-expression and lipid deposition (Figure 4). Here again, immunostaining for ABCA1 and ABCG1 was not performed.

In the paper, it is implied that lack of ABCA1/ABCG1 significantly impairs the recycling of lipids of phagocytosed OS. However, phagocytosis does not appear to have been tested in these conditional mice. Is the uptake/internalization of the phagocytosed OS normal in these mutated RPEΔAbca1;Abcg1 mice?

A decline in the expression of ABCA1 mRNA level was observed in the aged eyes (Supplementary figure S5). This finding would be more impactful if the tested samples would have been genotyped for the known AMD-associated ABCA1 risk-variant. What if the older donor eyes carry the risk-variant? The authors attempted to address the risk-genotype effect by showing a reduced level of the ABCA1 mRNA in the LCLs from individuals with ABCA1 risk-variant compared to control cells upon stimulation with an LXR agonist (Figure 11). In this experiment, reduction in expression of the ABCA1 mRNA (Figure 11A) in the risk-genotype cells did not correlate with the protein levels (Figure 11C). Also, these LCLs are quite different than RPE cells morphologically and functionally. The human data appear to be inconclusive and require additional experiments.

Reviewer #3

The authors targeted deletion of ABCA1 and its partner ABCG1 in mouse RPE, which resulted in retinal degeneration and inflammation, substantial loss of function of RPE and photoreceptors/neural retina, and accumulation of lipids in RPE (due to loss of efflux capacity). Assessment of human-derived cell lines from aged control and AMD patients confirmed increased AMD risk associated with decreased ABCA1 gene expression. The results highlight the importance of a functional ABC efflux machinery in the RPE for maintaining normal lipid homeostasis as well as retinal/RPE structure and function. Further, the results appear to have relevance to the pathobiological mechanism underlying AMD-- with the obvious caveat that the mouse has no macula and the life span is severely compressed relative to humans.

Overall, this is an elegant, well-conceived and technically well-executed study. The data (which were expansive in scope!) are rigorous and statistically sound, the manuscript is clearly written, and the quality of the data presentation is extremely good. A veritable tour de force!

This reviewer has no substantive concerns.

1) Anytime one meddles with a cell's genetic material, such as the targeted deletion performed in this study, there tends to be a "ripple effect"-- with expression of other genes being affected. The ABC family of transporters is complex, and integrates into a network of inter-connected gene families that regulate multiple aspects of cellular metabolism and homeostasis. It would be of interest to consider what other genes, besides ABCA1 and ABCG1, were affected (up/down expression of transcripts) by this targeted deletion-- both in RPE as well as in the neighboring neural retina. While this would be a rather time-consuming and expensive undertaking (so this reviewer is NOT requiring this for revision), the authors should at least consider the possibility of network effects in their discussion of the results.

2) While the comparison is often made to the cardiovascular system when discussing AMD and lipid homeostasis, it occurs to this reviewer that there's another parallel with the pulmonary system in the context of inflammation, disease, ABC transporters, and lipid homeostasis. A recent review highlights this: see Chai, Ammit and Gelissen, 2017.

Perhaps the authors would want to take this into consideration in their Discussion section as well?

---

## [Author Response]

The major item in our assessment is that we think that the post-mortem human data in Supplementary figure S5 are not compelling. One can see the issue by considering what the best fitting line would look like if the data point from the individual who died at ~age 30 is removed – the best fitting line would then be nearly horizontal. Our strong recommendation is to remove these data from the manuscript.

We absolutely agree that this data set is not very strong because only few young donors were available for testing. This misbalanced age distribution was the reason why we placed this data as a supplementary figure. To avoid presentation of potentially misleading data, we removed (old) Supplementary figure S5 and (old) Supplementary table S4 from the manuscript. However, we still mention in the Discussion section as ‘data not shown’ that preliminary experiments showed a tendency of reduced *ABCA1* expression in human RPE with age.

A second recommendation is to immunostain WT vs KO mouse RPE for the transporters to assess the KO at the protein level. In the manuscript, the analysis of Cre expression is nice but it is one step removed from the transporter. Also, the PCR analysis of the recombined fragment does not assess the fraction of cells in which recombination did not occur.

This is of course a very valid (and expected) comment. We invested a significant amount of time and effort and used several approaches to show downregulation of ABCA1/ABCG1 at the protein level. For Western blotting, we used two different buffers to isolate proteins from RPE-enriched cell isolates and whole eyecup preparations. All attempts failed to conclusively show reduced ABCA1 or ABCG1 protein levels. The only explanation we have is that anti-ABCA1 and anti-ABCG1 antibodies recognize epitopes outside the excised regions (epitope for ABCA1 lies within exons 23-29 (before the excised exons 45-46); epitope for ABCG1 is within exon 5 (after the excised exon 3)) and that (non functional) proteins are translated that are detected by the antibodies despite the excision of the respective floxed sequences. Such proteins should be smaller in size but since the proteins are very large (ABCA1: 254 kDa; ABCG1: predicted 75.5 kDa, runs around 110 kDa on Western blots), the size difference may not be easily detectable, especially since both antibodies proofed to perform poorly in Western blotting. In addition, *Cre* expression is patchy with many cells expressing full-length proteins, making it difficult to detect a difference by Western blotting. The presence of the flox/- genotype in a subset of cells (possibly in a chimeric fashion) in both Ctr and RPE*^∆Abca1;Abcg1^* mice can also influence the amount of protein, as it clearly influences the mRNA levels of *Abca1* and *Abcg1* (Figure 1—figure supplement 1B), and make the detection of a rather small difference even more difficult.

To circumvent some of these problems, we used immunofluorescence on retina sections and RPE flat mounts. In our hands, however, it was difficult to get meaningful intracellular staining of ABCA1/ABCG1 that discriminated between control and RPE*^∆Abca1;Abcg1^* mice, despite several attempts with different protocols. On flat mounts, no obvious differences were detectable, likely because of the production of truncated proteins recognized by the antibodies (see above).

Unfortunately, all of our many attempts showed only a tendency of reduced protein levels at most. We are very confident, however, that our Cre-LoxP system resulted in non-functional ABCA1/ABCG1 proteins at least in some RPE cells. With the presentation of the successful excision at the genomic DNA level (Figure 1), the specific *Cre* expression in the RPE (Figure 1), the lack of a phenotype in the *BEST1Cre* mouse (Figure 3—figure supplement 2), and the reduced mRNA levels (Figure 1—figure supplement 2) together with the strong RPE-specific phenotype of the RPE*^∆Abca1;Abcg1^* mice that fits well with reduced ABCA1/ABCG1 function, we believe that it is highly likely that function of ABCA1/ABCG1 was affected in our RPE-specific knockout mice. Moreover, the *Abca1^flox/flox^;Abcg1^flox/flox^* mice were previously used to successfully generate KO models in different tissues with a clear functional phenotype (Westerterp et al., 2012; Westerterp et al., 2016; Ban et al., 2018; Ban et al., 2018).

Of course, we are happy to provide all data of our attempts to demonstrate reduced ABCA1/ABCG1 protein levels upon request.

A third recommendation – and this one is less critical than the first two – is to conduct RNAseq on the WT vs KO RPE to see if there are secondary gene expression changes in response to transporter KO. Such changes, if they exist, could by informative.

Again, we completely agree with this recommendation. We asked this question ourselves and conducted an RNAseq experiment at 2 months of age to detect potential changes in the gene expression pattern of RPE*^∆Abca1;Abcg1^* mice. To do so, we determined the transcriptomes in retinas and eyecups of 4 control and 4 RPE*^∆Abca1;Abcg1^* mice. We did not include these data in the original manuscript because no major differences were observed that were of particular interest. Furthermore, samples did not cluster according to their genotype (see Author response image 1) suggesting that no major changes in the gene expression profile were induced by the knockout of the lipid transporters. With very few exceptions, differences in RNA levels as detected by RNAseq were only minor. Several of the differentially expressed genes were tested in independent samples by RT-PCR but none could be verified to be differentially affected by the knockout.

We now include a statement about these data in the Discussion section.

**Author response image 1. respfig1:** RNAseq data. Clustering of RNAseq data. Retinas (left) and eyecups (right) of 4 control and 4 RPE^∆*Abca1;Abcg1*^ mice at 2 months of age were subjected to RNAseq analysis. Samples did not cluster according to their genotype (Cre-positive or Cre-negative).

I am including the three reviews at the end of this letter, as there are a variety of specific and useful suggestions in them. We appreciate that the reviewers' comments cover a range of suggestions for improving the manuscript. We look forward to receiving your revised manuscript.Reviewer #1This study demonstrates the importance of the ABCA1-mediated lipid efflux pathway in maintaining RPE and photoreceptor cell health. Specifically, they show unequivocally that lack of ABCA1 in RPE cells resulted in lipid droplet accumulation within the RPE which, over time, led to loss of the RPE cells. The authors generated and characterized an RPE-specific Abca1;Abcg1 double knockout mice on a BEST1Cre background, they used 2 month old mice to show evidence of Cre mRNA expression in the eyecup and very little in the retina, IF positive staining for Cre only in the RPE nuclei, and showed CRE-mediated excision fragments of Abca1 and Abcg1 in the eyecups with some variability, likely due to the variable Cre expression in the RPE of these mice. Also, at 2 months of age, on a normal diet, morphologically there were already changes: a dotted pattern on fundus images, changes seen in the apical border of the RPE and lipid droplets in the RPE, lipidTOX and OilRedO staining coinciding with Cre expression in the RPE. A decrease in the rates of rhodopsin regeneration after bleaching demonstrated that this was slower initially in the double knockout mice suggesting impairment of visual cycle even in these 2 month old mice.Age-dependent changes were also seen by studying the mice at 2, 4 and 6 months. They detected deteriorating phenotypic changes by careful measurements of RPE shape, size, and Mct3 expression, light microscopy and EM showed increasing damage to the RPE and neural retina and increasing visual deficits by ERG measurements of 2, 4, and 6 month old mice.They used the correct controls to ensure changes seen in the mice were not developmental, they injected an AAV virus expressing CRE and GFP into adult Abca1flox/flox:Abcg1flox/flox mice and they saw, in regions that were GFP-positive, the same changes in accumulation of lipids and RPE damage they saw with the double knockout mice. To ensure the phenotype of the double KO was not due to CRE toxicity the BEST1CRE mouse was used as a control. They also generated RPE-specific Abca1 KO and RPE-specific Abcg1 KO mice and studying these at 2 months of age determined that the Abca1 appeared to be the dominant gene responsible for lipid efflux in the RPE cells, resulting in lipid accumulation in the cells, this may imply a somewhat different regulation of lipid homeostasis in RPE cells compared to macrophages that require both Abca1 and Abcg1.Overall a careful study which does not overstate the findings but nicely points to the ABCA1-mediated efflux of lipids as being essential for maintaining RPE health.It is unclear which of the analyses shown in Figure 5 were from "RPE-enriched samples", i.e. from an eyecup without the neural retina, to which PBS was added and then the tubes were flicked to release RPE cells and which analysis used the entire eyecup without the neural retina. Were all the analyses done on single eyes or were the tissue lysates pooled?

We apologize for the confusing description. The analyses for UC, CEs and REs were done separately from the rest of the lipid classes. We changed the labeling of the figure to reflect which analyses have been done on ‘RPE-enriched eyecup’ and which were done on ‘whole eyecup’ samples. This labeling is now also explained in the Materials and methods section and in the legend to Figure 5. Also, pooled tissues from both eyes of the same animal were used for UC, CEs and REs, while single eyes were analyzed for PLs, SL, and GLs. This is now explained in the Materials and methods section and in the legend to Figure 5.

In Figure 9A using 4 month old mice and flat mounts, sub-RPE IBA-1 positive cells infiltrating the RPE from the choroidal side are shown in flatmounts, can you give an indication of how many cells were seen and if you saw this in all the KO mice at this age.

We thank the reviewer for raising this very good point. After careful inspection of the images, we re-considered our initial interpretation of the results: since we cannot exclude the possibility that IBA-1-positive cells infiltrated the RPE from the apical side and migrated through the layer to the basal side, we now discuss the observed localization of the cells rather than speculating on the direction of their infiltration. All the flat mounts analyzed from RPE^∆*Abca1;Abcg1*^ mice at 4 months of age showed the presence of inflammatory cells in varying amounts: we now provide a rough estimate of the amount of IBA-1-positive cells per flat mount in the text. As we did not perform a staining for apical versus basal RPE markers and the mutant RPE is occasionally already damaged at this time point, it is often not possible to assign a clear localization (apical, middle or basal) to the IBA-1-positive cells in the RPE layer. The corresponding Results section was changed accordingly.

Then IBA-1 cells in 6 month mice showed, in retinal sections, sub-retinal immune cells were seen- were these seen in parts of the retina that still had an intact RPE layer or, as was seen in EM of sections of mice at this age had the RPE completely atrophied and disappeared. In the text, it appears there is a comparison made between the localization of immune cells at 4 months and 6 months, but because the figures show flat mounts for one time point and cross sections for the other the comparison is a little difficult to interpret.

We did not intend to compare the two time points because quantification of these cells is difficult due to variability in their shape and origin (blood-born macrophages versus activated retinal microglia). The main point from the IBA-1 immunofluorescence images is that we detect IBA-1 positive cells already at 4 months and also later at 6 months of age. We changed the text (subsection “Inflammatory response in RPE^Δ*Abca1;Abcg1*^ mice”) to make this point clearer and to state that IBA-1 positive cells were not only detected in the most affected retinal regions.

In Supplementary figure S5 using eyecup tissue from 25 human donor eyes, the authors attempted to show that there was a decrease in ABCA1 mRNA expression levels with age. Only 4 of these samples were from donors younger than 53 and without these 4 samples there did not appear to be any decline in ABCA1 expression from 57 to 96 years of age. This was the least convincing part of this project and slightly detracted from the rest of the work.

We absolutely agree that this data set is not very strong because only few young donors were available for testing. This misbalanced age distribution was the reason why we placed this data as a supplementary figure. To avoid presentation of potentially misleading data, we removed (old) Supplementary figure S5 and (old) Supplementary table S4 from the manuscript. However, we still mention in the discussion as ‘data not shown’ that preliminary experiments showed a tendency of reduced *ABCA1* expression in human RPE with age.

Introduction: Change 'choroids' to choroidDiscussion section: Replace 'phagocyte' with 'phagocytize'Discussion section: Replace 'synthetize' with 'synthesize'Discussion section: Change 'In presence' to 'In the presence'

All of the above has been changed in the text according to the recommendation of the reviewer.

Reviewer #2The manuscript by Storti et al., includes the characterization of a conditional knockout mouse that lacks ABCA1 and its effector protein ABCG1 in the RPE cells. RPEΔAbca1;Abcg1 mice presented an early onset ocular phenotype including lipid deposition in the RPE, local inflammatory reactivity, and progressive loss of both RPE and photoreceptor cells. Lymphoblastoid cells lines (LCL) from individuals with known AMD-risk variant of ABCA1 showed reduced ABCA1 expression when stimulated with a liver x receptor (LXR) agonist, further suggesting the role of ABCA1 in lipid metabolism. The manuscript is well-written for most of the parts. The study is more descriptive rather than providing new mechanistic insights on AMD. Given the similarities to AMD in the phenotypic features, the RPEDAbca1;Abcg1 mouse may be useful to test various therapeutic approaches targeting intracellular lipid/cholesterol metabolism. However, a major flaw of the study is the lack of direct investigations of ABCA1 and ABCG1 protein levels in the RPEΔAbca1;Abcg1 mouse by immunoblotting and/or immunohistochemistry (IHC).The ABCA1 and ABCG1 protein levels were evaluated by IHC only in the wild-type sections (Figure 1A). For the RPEΔAbca1;Abcg1 mouse, the read out for gene knockdown was evidenced primarily by Cre mRNA expression (Figure 1B and Figure 3F; Figure 1—figure supplement 2 and Figure 3—figure supplement 3) and at the protein level by IHC using an antibody against Cre rather than ABCA1 and ABCG1. Expression of Abca1 and Abcg1 mRNA (Figure 1—figure supplement 1 and Figure 1—figure supplement 2) was reduced in the mutated mouse but no protein data is shown.

We made many attempts (including Western blotting using different methodologies and protocols, immunofluorescence on sections and immunofluorescence on flat mounts) to show the effect of the CRE-mediated deletions on the protein levels. However, none of these attempts was convincing and we decided not to show the data. For more specific comments please see our reply to the second recommendation made by the Editor.

With the presentation of the successful excision at the genomic DNA level (Figure 1), the specific *Cre* expression in the RPE (Figure 1), the lack of a phenotype in the *BEST1Cre* mouse (Figure 3—figure supplement 2), and the reduced mRNA levels (Figure 1—figure supplement 2) together with the strong RPE-specific phenotype of the RPE*^∆Abca1;Abcg1^* mice that fits well with reduced ABCA1/ABCG1 function, we believe that it is highly likely that function of ABCA1/ABCG1 was affected in our RPE-specific knockout mice. Moreover, the *Abca1^flox/flox^;Abcg1^flox/flox^* mice were previously used to successfully generate KO models in different tissues with a clear functional phenotype (Westerterp et al., 2012; Westerterp et al., 2016; Ban et al., 2018; Ban et al., 2018).

Cre-driven morphological changes and increased lipid deposition in the RPE cells RPEΔAbca1;Abcg1 mouse were clearly evidenced in Figure 2 and Figure 5. The group also choose an AAV-based approach to transduce the RPE cells of Abca1 and Abcg1 Floxed/Floxed mouse to further draw a correlate between the GFP-Cre-expression and lipid deposition (Figure 4). Here again, immunostaining for ABCA1 and ABCG1 was not performed.

Please see our response to the point above. The same difficulties to detect levels of ABCA1 and ABCG1 on the protein level apply here.

In the paper, it is implied that lack of ABCA1/ABCG1 significantly impairs the recycling of lipids of phagocytosed OS. However, phagocytosis does not appear to have been tested in these conditional mice. Is the uptake/internalization of the phagocytosed OS normal in these mutated RPEΔAbca1;Abcg1 mice?

This is a very interesting point that we also considered but did not test directly *in vivo*. Based on the morphology of POS, the lack of obvious debris accumulation in the sub-retinal space (e.g. Figure 2), the accumulation of fatty acids typical for photoreceptor OS such as DHA in the RPE (Figure 5, see Discussion section) and the staining pattern obtained with an antibody against rhodopsin (not shown) in RPE*^∆Abca1;Abcg1^* mice, we have no evidence that uptake/internalization of shed OS was impaired in our mice.

Nevertheless, we generated three ARPE19 cell lines that stably expressed different anti-*ABCA1* shRNAs to investigate this in some more detail. Expression of *ABCA1* in these cell lines was at about 25% of control cells. The knockdown cells were fed with bodipy-cholesterol labeled OS from pig eyes and phagosomes were quantified at different time points after feeding. With this approach we detected a tendency towards reduced phagosome numbers at some but not all time points. Since number and size of cells at confluency strongly varied between these stable knockdown cell lines and control ARPE19 cells, data were not conclusive. No differences in phagosome numbers were detected when the same experiment was done with ARPE19 cells treated with anti-*ABCA1* siRNAs (transient knockdown).

Since phagocytosis was not at the heart of our research focus, we decided to abandon this line of experiments.

A decline in the expression of ABCA1 mRNA level was observed in the aged eyes (Supplementary figure S5). This finding would be more impactful if the tested samples would have been genotyped for the known AMD-associated ABCA1 risk-variant. What if the older donor eyes carry the risk-variant? The authors attempted to address the risk-genotype effect by showing a reduced level of the ABCA1 mRNA in the LCLs from individuals with ABCA1 risk-variant compared to control cells upon stimulation with an LXR agonist (Figure 11). In this experiment, reduction in expression of the ABCA1 mRNA (Figure 11A) in the risk-genotype cells did not correlate with the protein levels (Figure 11C). Also, these LCLs are quite different than RPE cells morphologically and functionally. The human data appear to be inconclusive and require additional experiments.

We agree with the reviewer that the effect of the risk-allele needs further investigation and that this data set is not very strong because only few young donors were available for testing. This misbalanced age distribution was the reason why we placed this data as a supplementary figure. To avoid presentation of potentially misleading data, we removed (old) Supplementary figure S5 and (old) Supplementary table S4 from the manuscript. However, we still mention in the Discussion section as ‘data not shown’ that preliminary experiments showed a tendency of reduced *ABCA1* expression in human RPE with age.

With regard to the data generated using LCLs from patients, we agree with the reviewer that these preliminary data (as we specifically labeled them in the manuscript) need to be confirmed. In particular, we will test iPSC-derived RPE cells from the patients to investigate cells that may resemble native RPE cells more closely. These cells will be used to test expression levels of ABCA1, and to analyze the potential influence of the risk allele on general metabolism and function. However, these experiments require more time and – to our feeling – are beyond the scope of this manuscript.

Reviewer #3The authors targeted deletion of ABCA1 and its partner ABCG1 in mouse RPE, which resulted in retinal degeneration and inflammation, substantial loss of function of RPE and photoreceptors/neural retina, and accumulation of lipids in RPE (due to loss of efflux capacity). Assessment of human-derived cell lines from aged control and AMD patients confirmed increased AMD risk associated with decreased ABCA1 gene expression. The results highlight the importance of a functional ABC efflux machinery in the RPE for maintaining normal lipid homeostasis as well as retinal/RPE structure and function. Further, the results appear to have relevance to the pathobiological mechanism underlying AMD-- with the obvious caveat that the mouse has no macula and the life span is severely compressed relative to humans.Overall, this is an elegant, well-conceived and technically well-executed study. The data (which were expansive in scope!) are rigorous and statistically sound, the manuscript is clearly written, and the quality of the data presentation is extremely good. A veritable tour de force!This reviewer has no substantive concerns.1) Anytime one meddles with a cell's genetic material, such as the targeted deletion performed in this study, there tends to be a "ripple effect"-- with expression of other genes being affected. The ABC family of transporters is complex, and integrates into a network of inter-connected gene families that regulate multiple aspects of cellular metabolism and homeostasis. It would be of interest to consider what other genes, besides ABCA1 and ABCG1, were affected (up/down expression of transcripts) by this targeted deletion-- both in RPE as well as in the neighboring neural retina. While this would be a rather time-consuming and expensive undertaking (so this reviewer is NOT requiring this for revision), the authors should at least consider the possibility of network effects in their Discussion of the results.

Again, we completely agree with this recommendation. We asked this question ourselves and conducted an RNAseq experiment at 2 months of age to detect potential changes in the gene expression pattern of RPE*^∆Abca1;Abcg1^* mice. To do so, we determined the transcriptomes in retinas and eyecups of 4 control and 4 RPE^∆*Abca1;Abcg1*^ mice. We did not include these data in the manuscript because no major differences were observed that were of particular interest. Furthermore, samples did not cluster according to their genotype (see Author response image 1) suggesting that no major changes in the gene expression profile were induced by the knockout of the lipid transporters. With very few exceptions, differences in RNA levels as detected by RNAseq were only minor. Several of the differentially expressed genes were tested in independent samples by RT-PCR but none could be verified to be differentially affected by the knockout.

We now include a statement about this data in the Discussion section.

2) While the comparison is often made to the cardiovascular system when discussing AMD and lipid homeostasis, it occurs to this reviewer that there's another parallel with the pulmonary system in the context of inflammation, disease, ABC transporters, and lipid homeostasis. A recent review highlights this: see Chai, Ammit and Gelissen, 2017.Perhaps the authors would want to take this into consideration in their Discussion section as well?

The function of ABCA1 and ABCG1 transporters in pulmonary lipid metabolism is indeed very interesting and results parallel our data. We incorporated the major finding of the Chai paper that a knockout of either gene results in a progressive and age-dependent lung phenotype, including lung dysfunction and inflammation in the Introduction.